# A new sea ice concentration product in the polar regions derived from the FengYun-3 MWRI sensors

Ying Chen[1], Ruibo Lei[2,1], Xi Zhao[3], Shengli Wu[4,5], Yue Liu [6,1], Pei Fan[1], Qing Ji[1], Peng Zhang[4,5], Xiaoping Pang[1†]

[1]Chinese Antarctic Center of Surveying and Mapping, Wuhan University, Wuhan, 430079, China
[2]Key Laboratory for Polar Science of the MNR, Polar Research Institute of China, Shanghai, China
[3]School of Geospatial Engineering and Science, Sun Yat-Sen University & Southern Marine Science and Engineering Guangdong Laboratory, Zhuhai, China
[4]Key Laboratory of Radiometric Calibration and Validation for Environmental Satellites, National Satellite Meteorological Center (National Center for Space Weather), China Meteorological Administration, Beijing, China
[5]Innovation Center for FengYun Meteorological Satellite (FYSIC), Beijing, China
[6]Jiangsu Provincial Surveying and Mapping Engineering Institute, Nanjing, China

[†]*Correspondence to*: Xiaoping Pang (pxp@whu.edu.cn)

**Abstract.** Sea ice concentration (SIC) is the main geophysical variable for quantifying the change in sea ice in the polar regions. A continuous SIC product is key to inform climate and ecosystems studies in the polar regions. Our study generates a new SIC product covering the Arctic and Antarctic from November 2010 to December 2019. It is the first long-term SIC product derived from the Microwave Radiation Imager (MWRI) sensors onboard the Chinese FengYun-3B, -3C, and -3D satellites, after a recent re-calibration of brightness temperature. We modified the previous Arctic Radiation and Turbulence Interaction Study Sea Ice (ASI) dynamic tie points algorithm mainly by changing input brightness temperature and initial tie points. The MWRI-ASI SIC was compared to the existing ASI SIC products and validated using ship-based SIC observations. Results show that the MWRI-ASI SIC mostly coincides with the ASI SIC obtained from the Special Sensor Microwave Imager series sensors, with overall biases of -1 ± 2% in the Arctic and 0.5 ± 2% in the Antarctic, respectively. The overall mean absolute deviation between the MWRI-ASI SIC and ship-based SIC is 16% and 17% in the Arctic and Antarctic, respectively, which is close to the existing ASI SIC products. The trend of sea ice extent (SIE) derived from MWRI-ASI SIC closely agrees with those of the Sea Ice Index SIEs provided by OSI-SAF and NSIDC. Therefore, the MWRI-ASI SIC is comparable with other SIC products and may be applied alternatively. The MWRI-ASI SIC dataset is available at https://doi.pangaea.de/10.1594/PANGAEA.945188 (Chen et al., 2022).

## 1 Introduction

The sea ice concentration (SIC) and extent (SIE), which have been continuously and regularly provided by passive microwave (PM) remote sensing for more than four decades (Trewin et al., 2021; Lavergne et al., 2022), are crucial phenological indicators for the changes in the climate and marine environment in the polar regions. The PM SIC is the most vital dataset to initialize the sea ice condition for climate modeling due to its continuous observations (Meier, 2019). The SIE in the polar regions has

significant annual cycles and year-to-year variations, which are closely related to the changes in climate and ecosystem in the polar regions, and global ocean circulation, suggesting significance as a climate index on both regional and global scales

(Comiso et al., 2017; Parkinson and DiGirolamo, 2021; Heil et al., 2006).

Due to the long-term services of spaceborne sensors, the PM measurements can be used to continuously track the response of sea ice to climate change and support the applications for climate models or multidisciplinary studies in the polar regions. There is a risk for the continuity of the PM timeseries as the current missions (Special Sensor Microwave Imager Sounder (SSMIS) and Advanced Microwave Scanning Radiometer 2 (AMSR2)) have long exceeded their design life (Gerland et al.,

2019). The new missions for successors of these two sensors or the launch plans for other instruments, e.g., the Copernicus Imaging Microwave Radiometer (Jiménez et al., 2021) and Weather Satellite Follow–On–Microwave (Newell et al., 2020), are in planning or concept stage (Esastar, accessed 2023-06-10). The Chinese instruments, the Microwave Radiation Imager (MWRI) sensors onboard the FengYun-3 (FY-3) series satellites, i.e., FY-3A, FY-3B, FY-3C, and FY-3D (Zhang et al., 2018, 2019; Xian et al., 2021), promise to provide independent long-term SIC (Chen et al., 2021; Zhao et al., 2022). However, the

inconsistency of different sensors and the drift of the sensor itself with increased operation time can increase the uncertainties of SIC (Eisenman et al., 2014). Thus, it is important to acquire consistent SIC and SIE data from various sensors. SSMIS data have been used to bridge the 2011 to 2012 data gap between Advanced Microwave Scanning Radiometer-EOS (AMSR-E) and AMSR2 (Meier and Ivanoff, 2017).

The SIE products derived from various sensors or algorithms revealed biases, ranging from $0.5\times10^6$ to $1\times10^6$ km$^2$, or about 3%

(in winter) to 20% (in summer) of the total Arctic or Antarctic SIE (Meier and Stewart, 2019). The main factors affecting the uncertainties of SIC and SIE are the sensitivities of SIC algorithms to atmospheric emission and sea ice emissivity, especially for thin or melting sea ice (Ivanova et al., 2015). The SIC products with higher spatial resolutions outperform when detecting ice edge but would underestimate SIE in the marginal ice zone (MIZ) because they tend to ignore thin or melting ice (Meier and Stewart, 2019). Thus, the uncertainties of SIC and SIE are generally greater during sea-ice melt rather than during freezing

conditions. The uncertainties of SIC and SIE are also due to the different operations to remove spurious sea ice caused by the weather effects and land spillover (Meier and Stewart, 2019; Kern et al., 2019).

The ship-based visual observations of sea ice are the main data used for the ground validation of the satellite-based PM SIC products. To assess the performance of various algorithms for the SIC products, Spreen et al. (2008) compared the AMSR-E SICs derived from the Arctic Radiation and Turbulence Interaction Study Sea Ice (ASI), Bootstrap (BST), and enhanced NASA

Team (NT2) algorithms to the ship-based SIC observations. Results indicated that the three SIC products were slightly lower than the ship-based SIC in winter but higher in summer by 10% to 12%, because the small-scale morphological features such as leads and sparse small floes are unresolved by the PM observations. Xie et al. (2013) compared the summer AMSR-E SIC to the ship-based SIC obtained in the Arctic Ocean and revealed that the AMSR-E SIC was overestimated in the pack ice zone (PIZ) but underestimated in the MIZ. Kern et al. (2019) evaluated 10 PM SIC products using the ship-based SIC observations

in the polar regions with medium SIC and revealed the SIC products derived from the BST algorithm had the lowest deviations against the ship-based SIC.

The first generation of daily SIC dataset derived from the MWRI sensors has been released by the Chinese National Satellite Meteorological Center (NSMC) in June 2011 using the NT2 algorithm, which had a considerable positive systematic deviation compared to the SIC obtained from the Interactive Multisensor Snow and Ice Mapping System, especially at the ice edge in summer (Wu and Liu, 2018). Due to low frequencies applied in the NT2 algorithm, the original resolutions of the NT2 SIC products are lower than those of the ASI SIC products, which only uses the highest frequency with high spatial resolution (Spreen et al., 2008). Zhao et al. (2022) produced a preliminary one-year Arctic SIC product derived from the FY-3D MWRI sensor using a dynamic tie points ASI algorithm, which had a smaller deviation against the AMSR2 SIC derived from the ASI algorithm compared to the products of Sea Ice Index SIC, OSI-430-b SIC, and AMSR2 SIC derived from the BST or NT2 algorithm.

In order to promote the application of MWRI sensors, this study extends the work of Zhao et al. (2022) and generates a new polar SIC product from November 2010 to December 2019. The recently re-calibrated brightness temperature (TB) of the MWRI sensors provided by NSMC are used in this study to ensure the consistency of this new MWRI SIC product. The previous ASI algorithm involving dynamic tie points is modified to obtain a longer MWRI SIC product. Moreover, the MWRI-ASI SIC is compared to the existing ASI SIC products and assessed systematically using ship-based SIC observation to identify its uncertainty in various regions and seasons. We also derive SIE from the MWRI-ASI SIC and compare it to the existing SIE products to test its potential for independent application.

## 2 Data and method

### 2.1 TB data from PM sensors

The MWRI sensors measure the radiation of the land, ocean, and atmosphere in conical scanning mode at five frequencies between 10 to 89 GHz at both horizontal (H) and vertical (V) polarization. The footprint size of the individual frequency ranges from 9 km at 89 GHz to 85 km at 10.65 GHz. Further details of the MWRI characteristics are given in Zhao et al. (2022). Although the MWRI sensors onboard the different FY-3 satellites have consistent technical characteristics, the TB data obtained from different MWRI sensors reveal some deviations. Therefore, the MWRI TB data are re-calibrated using the operational algorithm, which focus on the hot load, antenna, and receiver calibration, reducing the TB deviations of different MWRI sensors.

This study uses the re-calibrated level 1 swath MWRI TB data from the FY-3B, FY-3C, and FY-3D satellites, provided by the NSMC and available at http://www.richceos.cn (Table 1). Considering the improved performance of the FY-3D MWRI sensor compared to its predecessors, we preferentially selected the MWRI TB from the FY-3D, followed by the FY-3C and FY-3B. Determined by the availability and quality of the MWRI TB, the FY-3B data covered the two periods from 12 November 2010 to 30 September 2013 and from 31 May to 10 July 2015; the FY-3C lasted from 1 October 2013 to 30 May 2015 and from 11 July 2015 to 31 December 2017; and the FY-3D covered two years of 2018 and 2019. The re-calibrated swath MWRI TB data

at 89 GHz with V- and H-polarization were applied for the ASI algorithm, and those at 18.7, 23.8, and 36.5 GHz with V-polarization were served for the weather filters. These five channels were projected onto a polar stereographic grid true at 70 degrees with a 12.5-km spatial resolution.

To evaluate the uncertainties of the re-calibrated MWRI TB in the polar regions, we chose two daily TB products, i.e., the SSMI TB (version 6, Meier et al., 2021) and AMSR TB (AMSR-E version 3, Cavalieri et al., 2014; AMSR2 version 1, Meier et al., 2018), which are both available from the National Snow and Ice Data Center (NSIDC). This SSMI TB product is projected on 12.5- and 25-km polar stereographic grids at high and low frequencies, respectively. All the frequencies of the AMSR TB products are projected on a 12.5-km polar stereographic grid. The temporal coverage of these two daily TB products is corresponding to that of the MWRI TB. To conduct a comparison among these three TB products, the swath MWRI TB was gridded to daily MWRI TB and the low frequencies of SSMI TB were resampled to the 12.5-km polar stereographic grid. The regional TB differences were calculated in the PIZ, MIZ, and open water, respectively.

**Table 1. Summary of the datasets of TB, sea ice surface melt/freeze onset, snow depth, SIC, and SIE used in this study.**

| Parameter | Dataset | Source | Available period | Sensor | Algorithm | Resolution (km) |
|---|---|---|---|---|---|---|
| TB | MWRI TB | NSMC | 11/2010 – 12/2019 | MWRI | - | 12.5 |
| | SSMI TB | NSIDC | 11/2010 – 12/2019 | SSM/I, SSMIS | - | 25 / 12.5 |
| | AMSR TB | NSIDC | 11/2010 – 10/2011 07/2012 – 12/2019 | AMSR-E AMSR2 | - | 12.5 |
| Melt/freeze | Melt/freeze onset | GESR | 2011 – 2019 | SSM/I, SSMIS | PMA | 25 |
| Snow depth | Snow depth | NSIDC | 11/2010 – 10/2011 07/2012 – 12/2019 | AMSR-E AMSR2 | snow-depth-on-sea-ice | 12.5 |
| SIC | SSMI-ASI | Hamburg Uni. | 11/2010 – 12/2019 | SSM/I, SSMIS | ASI | 12.5 |
| | AMSR-ASI | Bremen Uni. | 11/2010 – 10/2011 07/2012 – 12/2019 | AMSR-E AMSR2 | ASI | 6.25 |
| SIE | SSMI-BST | NSIDC | 11/1978 – 12/2019 | SMMR, SSM/I, SSMIS | BST | 25 |
| | SSMI-NT | NSIDC | 11/1978 – 12/2019 | SMMR, SSM/I, SSMIS | NT | 25 |
| | Sea Ice Index | NSIDC | 11/1978 – 12/2019 | SMMR, SSM/I, SSMIS | revised NT | 25 |
| | OSI-SAF | OSI-SAF | 11/1978 – 12/2019 | SMMR, SSM/I, SSMIS | Bristol & BST | 25 |

## 2.2 Existing ASI SIC products

Two daily SIC products using the ASI algorithm in the polar regions were used as comparative data in this study (Table 1). One is available from the Integrated Climate Data Center (ICDC) of the University of Hamburg, which is derived from the Special Sensor Microwave Imager series projected onto a 12.5-km polar stereographic grid (SSMI-ASI) (Kern et al., 2023). The other is derived from the Advanced Microwave Scanning Radiometer series projected onto a 6.25-km polar stereographic grid (version 5.4, AMSR-ASI) (Melsheimer and Spreen, 2023). For comparison during the overlap periods with the MWRI data, we used the SSMI-ASI SIC from November 2010 to December 2019, the AMSR-E ASI SIC from November 2010 to

October 2011, and the AMSR2 ASI SIC from July 2012 to December 2019, respectively. Comparisons of the daily SICs and SIEs (SIC > 15%) derived from the MWRI-ASI, SSMI-ASI, and AMSR-ASI were performed at their native spatial resolutions.

To evaluate differences in the uncertainties of SIC between the melting and freezing periods, we use the data of Arctic sea ice surface melt or freeze onset to define the ice melting and freezing periods, which (version 371s, Table 1) are available from the Goddard Earth Science Research (https://earth.gsfc.nasa.gov/index.php/cryo/data). These data are obtained from the SSMI series sensors using the passive microwave algorithm (PMA) projected onto a 25-km polar stereographic grid, which includes the onsets of the early melt, melt, freeze, and late freeze for the sea ice surface (Markus et al., 2009). We resampled the three ASI SIC products onto a 25-km grid to keep consistent with these data.

To quantify the effects of snow depth on SIC uncertainties, we obtained the snow depth on sea ice for the Arctic and Antarctic from the NSIDC (Table 1, AMSR-E version 3, Cavalieri et al., 2014; AMSR2 version 1, Meier et al., 2018). These data are derived from the AMSR TB using the AMSR-E snow-depth-on-sea-ice algorithms and projected on a 12.5-km polar stereographic grid. It is noted that these data are averaged by a five-day running window and only includes the depth of dry snow. These data provide snow depth for the entire South Ocean in the Antarctic, but only for the first-year ice in the Arctic.

To assess the performance of different SIC products in the MIZ, where the accuracy of PM SIC is generally low, the monthly MIZ SIE and MIZ SIE fraction (the ratio between the MIZ SIE and the total SIE) obtained from the three ASI SIC products are compared. We resample the AMSR-ASI SIC onto a 12.5-km grid to match the MWRI-ASI SIC and SSMI-ASI SIC to compare the SIC for the entire instrument overlap, as well as the winter (Arctic: December – May, Antarctic: June – November) and summer months (Arctic: June – November, Antarctic: December – May), respectively. To evaluate the uncertainties of SIC under different SIC scenarios, we divided SIC into three levels: low SIC (15–30%), medium SIC (30–70%), and high SIC (70–100%), and calculated the SIC differences among the ASI SIC products within each SIC category.

## 2.3 Existing monthly SIE products

This study used monthly SIE from four satellite-derived products in the polar regions from November 1978 to December 2019 (Table 1), which are all derived from SIC products at a 25-km grid resolution obtained from the SSMI series sensors. The NSIDC provides the SIE products using the BST and NASA Team algorithms (SSMI-BST and SSMI-NT) (Stroeve and Meier, 2018), as well as the SIE product derived from the Sea Ice Index SIC product (version 3) (Fetterer et al., 2017). The fourth SIE product (called as OSI-SAF) is derived from the SIC product described in Lavergne et al. (2020).

The differences between the MWRI-ASI SIE and four existing SIE products are quantified for the entire instrument overlap, as well as the winter and summer months. The 2010–2019 trends of the MWRI-ASI SIE and four existing SIE products are compared. To validate the capability of the MWRI-ASI SIE for climate and ecosystems studies, we performed an analysis of combined SIE trends. The 40-year (1979 to 2019) trends combining the four existing SIE products from January 1979 to November 2010 and MWRI-ASI SIE from December 2010 to December 2019 were compared to the original trends derived from the four existing SIE products.

## 2.4 Modified ASI dynamic tie points algorithm

In a previous study (Zhao et al., 2022), a TB bias-correction was performed to reduce the bias between daily MWRI TB and AMSR2 TB. An ASI algorithm involving daily dynamic tie points was applied in the Arctic. The tie points of AMSR series were used as the initial tie points, and the daily dynamic tie points were generated according to the initial SIC, locations, and time sliding window. Two weather filters, i.e., GR(36.5/18.7) and GR(23.8/18.7), and a monthly maximum ice extent mask were utilized to remove the spurious sea ice. The gradient ratio (GR) is defined as the TB difference with V-polarization between

high and low frequencies over the sum of these two TB. Details about the dynamic tie points ASI algorithm are given in Zhao et al. (2022), and the details about the ASI algorithm can be referred to Svendsen et al. (1987), Kaleschke et al. (2001) and Spreen et al. (2008).

    To obtain a longer dataset of MWRI SIC and optimize the estimation procedures, this study modified the previous algorithm from the five aspects (Table 2). The detailed procedures for retrieving this MWRI-ASI SIC product can be seen in the

supplement file. Using daily TB would dilute the atmospheric signal due to the nonlinear atmospheric influence on the TB (Comiso et al., 2003). Thus, this study used the re-calibrated swath MWRI TB to calculate SIC and gridded the swath SIC into daily SIC.

**Table 2. Parameters or operations used in the previous and modified algorithms.**

| Parameter/operation | Previous algorithm | Modified algorithm |
|---|---|---|
| input TB | daily MWRI TB bias-corrected to daily AMSR2 TB | swath re-calibrated MWRI TB |
| swath into daily | swath TB into daily TB | swath SIC into daily SIC |
| initial tie points | $P_1$ =11.7 K, $P_0$ = 47 K (Arctic) | $P_1$ =7.1 K, $P_0$ = 50.3 K (Arctic) <br> $P_1$ =7.3 K, $P_0$ = 55.9 K (Antarctic) |
| dynamic tie points (Antarctic) | - | $P_1$: initial SIC larger than 95% within the monthly minimum ice extent, 100 km away from the coast. <br> $P_0$: initial SIC within [-10%, 10%], between 200 and 350 km away from the monthly ice edge, 100 km away from the coast. |
| weather filters | GR(36.5/18.7): 0.045; GR(23.8/18.7): 0.04 | GR(36.5/18.7): 0.05; GR(23.8/18.7): 0.045 |

Zhao et al. (2022) directly used the tie points from the AMSR series to initiate the SIC derivation, which gives rise large bias in initial SIC due to differences between MWRI and AMSR TB. To overcome this deficiency, our study applied a new operation to generate the initial tie points. We tested the sea ice tie points ($P_1$) from 6.0 to 12.0 K and the open water tie points ($P_0$) from 47.0 to 57.0 K with an interval of 0.1 K. Based on these 6000 pairs of tie points, the swath MWRI SIC was calculated from the swath TB using the ASI algorithm and then averaged into daily SIC. We used daily SSMI-ASI SIC in 2018 as the

reference SIC and computed the daily average of the MWRI-ASI SIC and SSMI-ASI SIC (SIC > 15%). The next step involves the linear regression between the daily average MWRI-ASI SIC and referential SSMI-ASI SIC. As initial tie point the pair satisfying requirements with the slope closer to 1, intercept closer to 0, and relatively low standard deviation (Std) was selected

from the 6000 samples. According to the above procedures, the initial $P_1$ was defined as 7.1 and 7.3 K, and the $P_0$ was defined as 50.3 and 55.9 K in the Arctic and Antarctic, respectively. This parameterization of $P_1$ and $P_0$ can effectively reduce the differences between the initial MWRI SIC and referential SSMI SIC. Details on how to generate the initial tie points are provided in Section S1.2.

We adopted the dynamic tie points proposed by Zhao et al. (2022) in Arctic, and process for Antarctic tie points is outlined next. Antarctic sea ice tie points are derived as: the initial SICs of grids are larger than 95%, and the grids are within the monthly minimum ice extent and 100 km away from the coast. Antarctic open water tie points are derived as: the initial SICs of grids fall within the range [-10%, 10%], and the grids are away from the monthly ice edge by 200–350 km and away from the coast by 100 km. Details on generating the dynamic tie points are given in Section S1.4.

The thresholds of the weather filters GR(36.5/18.7) and GR(23.8/18.7) are determined as 0.045 and 0.04 by Zhao et al. (2022), respectively, as the AMSR series sensors. In this study, we chose 0.05 and 0.045 as thresholds of GR(36.5/18.7) and GR(23.8/18.7), respectively, as the SSMI series sensors, which generally remove the weather effects (Gloersen and Cavalieri, 1986; Cavalieri et al., 1995).

## 2.5 Ship-based observation data

The ship-based observations of sea ice follow the protocols of the Ice Watch/Arctic Ship-based Sea-Ice Standardization (Ice Watch/ASSIST) (Hutchings et al., 2019) in the Arctic and of the Antarctic Sea Ice Processes and Climate (ASPeCt) (Worby and Allison, 1999) in the Antarctic. The Arctic protocol builds on the Antarctic one by adding specific surface description such as melt pond coverage. To validate the accuracy of the MWRI-ASI SIC, we collected the observational SIC from various ship-based measurement programs of the Chinese National Arctic and Antarctic Research Expedition (CHINARE) conducted by the Polar Research Institute of China (Lei et al., 2017) and a standardized ship-based observation dataset (ESA-SICCI) produced by Kern (2019), as well as those available in the IceWatch (https://icewatch.met.no/cruises) and PANGAEA databases (https://www.pangaea.de) (Arndt, 2019; Arndt and van Caspel, 2017; Katlein et al., 2014; Arndt, 2018; Arndt and Castellani, 2019; Hendricks et al., 2012).

A total of 8887 and 3882 ship-based observations in the Arctic and Antarctic, respectively, obtained from December 2010 to November 2019 were used here. Among them, 10726 and 2043 samples were obtained from the summer and winter months, respectively. We projected the ship-based SIC onto the polar stereographic grid and computed the average of the ship-based samples obtained in one calendar day within one polar stereographic grid corresponding to the PM SIC sample. A total of 5230, 5508, and 6169 samples of the ship-based SIC corresponding to the MWRI-ASI, SSMI-ASI, and AMSR-ASI products were used in the Arctic, respectively, about 88% (12%) of which were obtained from the summer (winter) months. In the Antarctic, we collected 2599, 2613, and 2979 ship-based SIC samples corresponding to the MWRI-ASI, SSMI-ASI, and AMSR-ASI products, respectively, about 73% (27%) of which were obtained from the summer (winter) months.

We compared the three ASI SIC products to the ship-based SIC by calculating the bias, mean absolute deviation (MAD), root mean standard deviation (RMSD), and correlation coefficient ($R$) during the entire overlap periods, as well as the summer and

winter months separately, at their native spatial resolutions. To evaluate the impact of sea ice conditions on the accuracy of SIC, we divided SIC as low (15–30%), medium (30–70%), and high (70–100%) levels.

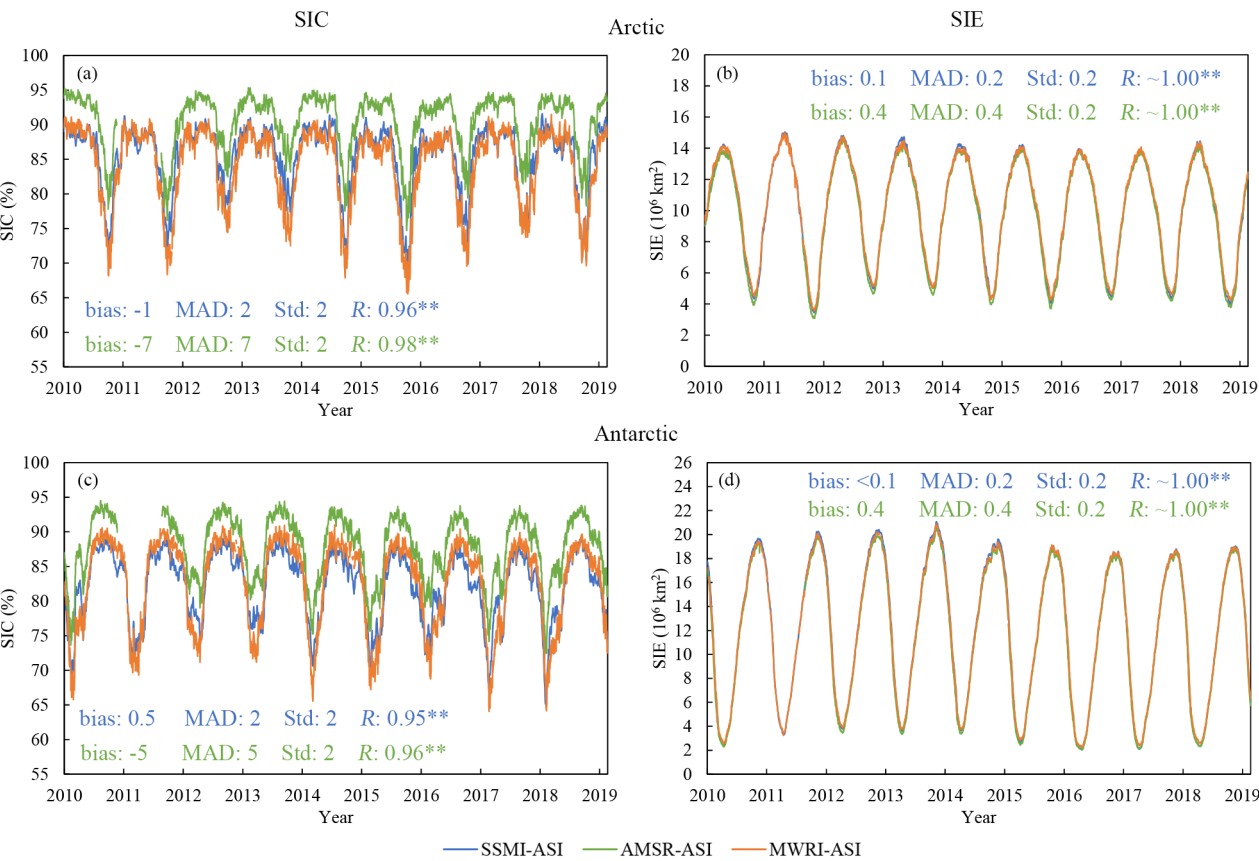

**Figure 1: Daily average SIC and daily SIE from November 2010 to December 2019. Also shown are the differences in SIC and SIE between the MWRI-ASI and SSMI-ASI (blue number) and between the MWRI-ASI and AMSR-ASI (green number). The statistical significance at 95% and 99% confidence levels are marked by * and **, respectively, and those below 95% confidence level are not marked, the same below.**

## 3 Results

### 3.1 Comparison of the ASI SIC products

All the three ASI SIC products reveal similar variation patterns in daily SIC and SIE (Fig. 1), with $R$s between each pair of products larger than 0.95 ($P<0.01$). The MWRI-ASI SIC is much closer to the SSMI-ASI SIC with overall biases of -1% and 0.5% in the Arctic and Antarctic, respectively, compared to the AMSR-ASI SIC. Both the MWRI-ASI SIC and SSMI-ASI SIC are lower than the AMSR-ASI SIC by about 5% in the bipolar regions. The MADs of SIE between the MWRI-ASI and SSMI-ASI are lower than those between the MWRI-ASI and AMSR-ASI by $0.2\times10^6$ km$^2$ (or 2% of the mean total SIE) in the

bipolar regions. Compared to the AMSR-ASI, the relatively small differences in SIC and SIE of the MWRI-ASI against the SSMI-ASI is likely because the initial tie points computation of the MWRI-ASI SIC was referred to the SSMI-ASI SIC.

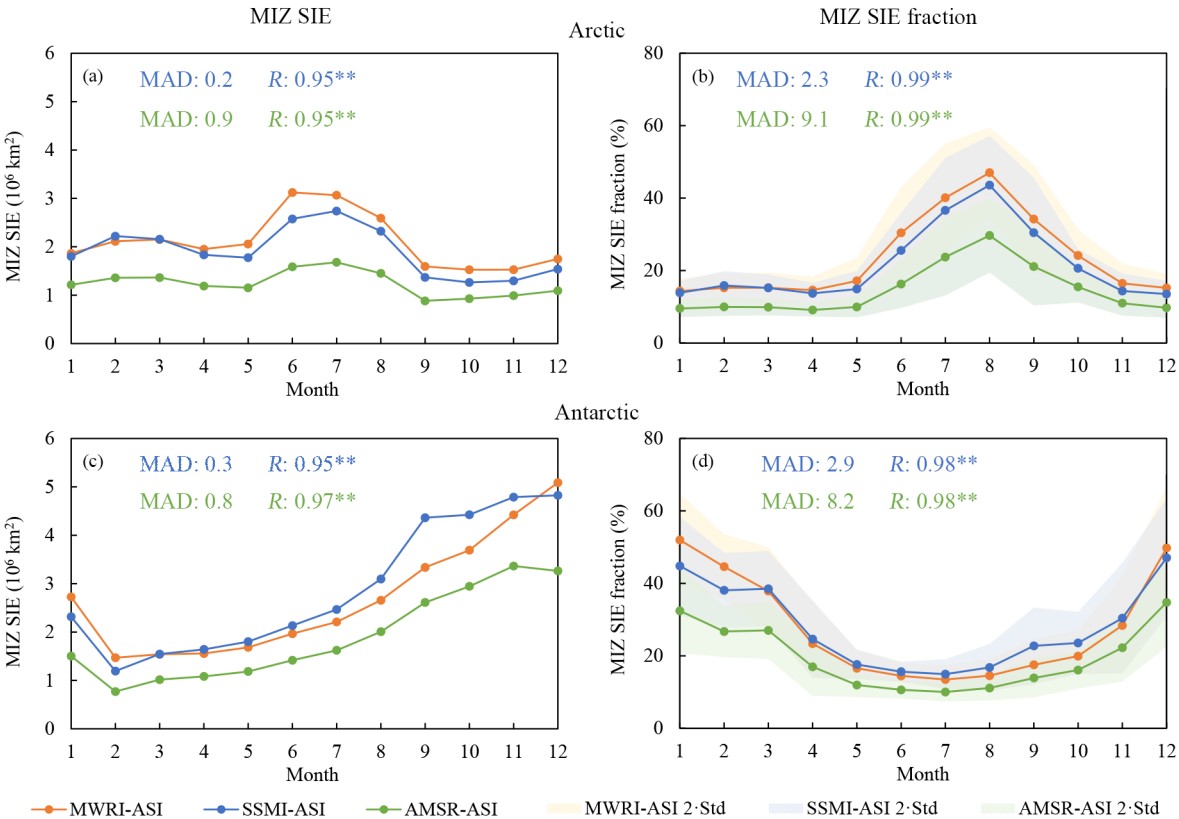

**Figure 2:** Monthly MIZ SIE and MIZ SIE fraction from November 2010 to December 2019. Also shown are the differences in MIZ SIE and MIZ SIE fraction between the MWRI-ASI and SSMI-ASI (blue number) and between the MWRI-ASI and AMSR-ASI

(green number). The shades present the 2 Stds from the monthly MIZ SIE fraction.

Compared to the total SIE, the MADs of MIZ SIE between the MWRI-ASI and SSMI-ASI increase by $0.04\times10^6$ and $0.1\times10^6$ $km^2$, and those between the MWRI-ASI and AMSR-ASI increase by $0.5\times10^6$ and $0.4\times10^6$ $km^2$ in the Arctic and Antarctic, respectively (Fig. 2). It suggests that the deviations of different SIC products increase significantly in the MIZ compared to the total ice region. Seasonally, the MWRI-ASI MIZ SIE reveals the smallest deviation against the SSMI-ASI MIZ SIE in

March for the bipolar regions. However, the influence regime is different between the Arctic and Antarctic. In the Arctic, fewer spurious ice along the coast caused by land spillover is introduced into the MIZ SIE estimation in March, when the SIE reaches the maximum, reducing the deviation between two SIEs. In the Antarctic, lower MWRI-ASI MIZ SIE is exactly compensated by the spurious ice caused by land spillover and weather effects in March, leading to a near-zero bias between two SIEs. In the bipolar regions, the differences in MIZ SIE between the MWRI-ASI and AMSR-ASI are lower in winter than in summer when the MIZ SIE fraction is higher. The MIZ SIEs obtained from the MWRI-ASI SIC and SSMI-ASI SIC are

larger than that obtained from the AMSR-ASI SIC, because lower grid resolutions of the MWRI-ASI and SSMI-ASI lead to

smearing at the ice edge and overestimation of MIZ SIE. Moreover, the MIZ SIE fraction of the AMSR-ASI is lower than those of the MWRI-ASI and SSMI-ASI, because the total AMSR-ASI SIC is relatively high (Fig. 1) and more grids at the boundary between the MIZ and PIZ have been identified as the PIZ by the AMSR-ASI compared to the MWRI-ASI and SSMI-
ASI.

Spatially, the MWRI-ASI SIC is slightly smaller than the SSMI-ASI SIC in the PIZ in the Arctic, while larger along the coastline and ice edge (Fig. 3), as the MWRI-ASI has more residual ice caused by land spillover and weather effects compared to the SSMI-ASI. In the Antarctic, the MWRI-ASI SIC is generally higher than the SSMI-ASI SIC. Compared to the AMSR-ASI SIC, the MWRI-ASI SIC is underestimated in the pan Arctic Ocean and in the PIZ in the Antarctic, while overestimated
at the ice edge in the Antarctic. Note that the apparent stripes of SIC differences between the MWRI-ASI and SSMI-ASI are located at the low latitude regions in the bipolar regions (Fig. 3a, 3b, 3g, and 3h), which are due to the raw stripes of the SSMI-ASI SIC.

In the Arctic and Antarctic, the lowest SIC differences both appear in the region with high SIC (70–100%) (Fig. 4), with mean MAD of 4% between the MWRI-ASI and SSMI-ASI and of 5% between the MWRI-ASI and AMSR-ASI, respectively. The
largest MAD (16%) between the MWRI-ASI SIC and SSMI-ASI SIC is identified in the region with low SIC (15–30%), and the MAD between the MWRI-ASI SIC and AMSR-ASI SIC is the highest (16%) in the region with medium SIC (30–70%) or low SIC (15–30%). Thus, the differences among the ASI SIC products are smaller in the regions of high SIC than in the regions with low or medium SIC.

In the Arctic, in the freezing period of sea ice surface from late freeze onset to early melt onset, the SIC differences are lower
with MAD of 3% between the MWRI-ASI and SSMI-ASI and of 4% between the MWRI-ASI and AMSR-ASI, respectively, compared to those (6% and 9%) during surface melt from melt onset to freeze onset. The MWRI-ASI SIC is higher than the SSMI-ASI SIC in the surface freezing stage but slightly smaller in the surface melting stage. These indicate that the differences among the ASI SIC products are larger in the surface melting state than in the surface freezing state and that the MWRI-ASI is more sensitive to melting ice surface than the SSMI-ASI.

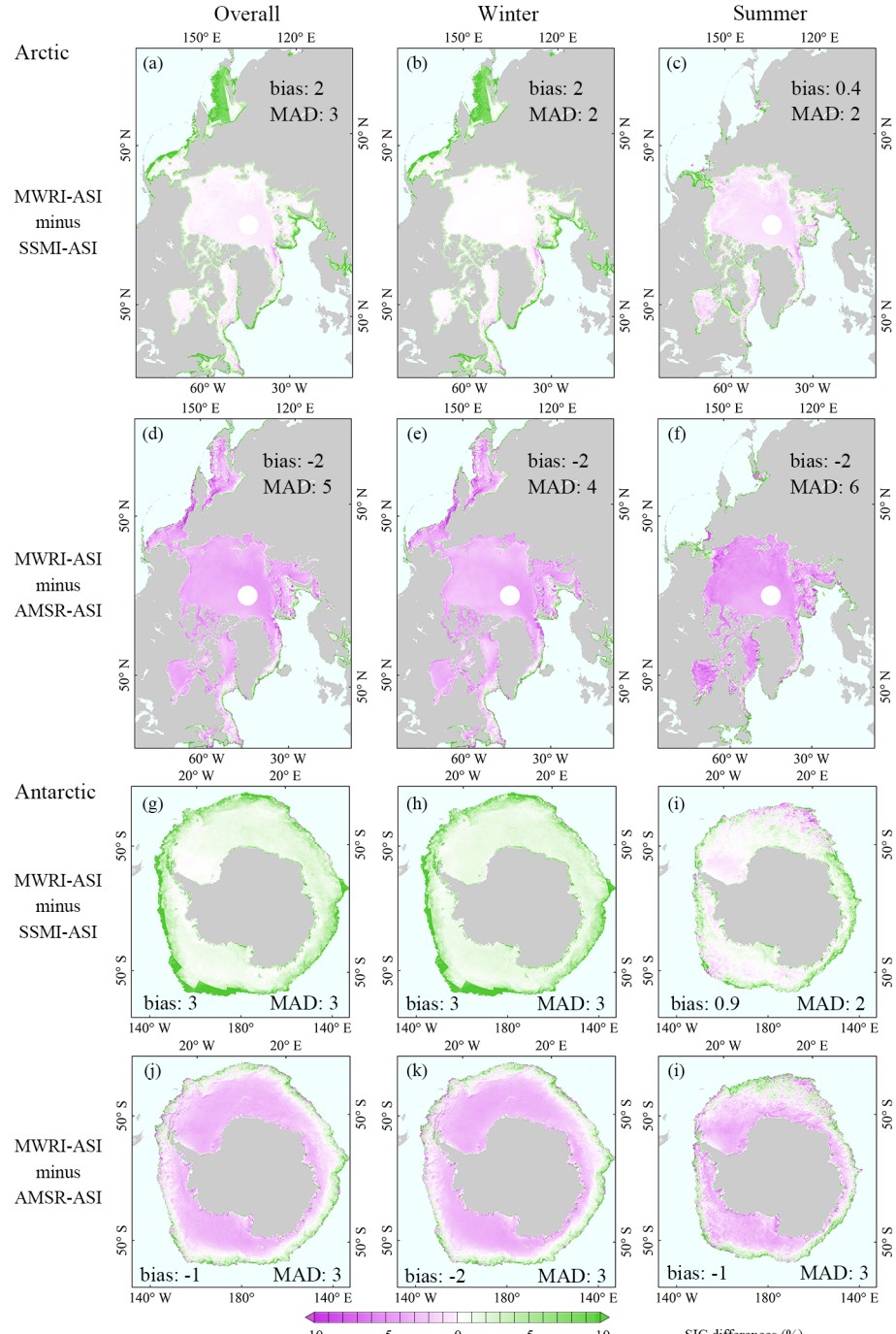

**Figure 3: SIC differences between the MWRI-ASI and SSMI-ASI and between the MWRI-ASI and AMSR-ASI for the entire time-series, as well as the winter and the summer months from November 2010 to December 2019. The grey is land, and the cyan is open water.**

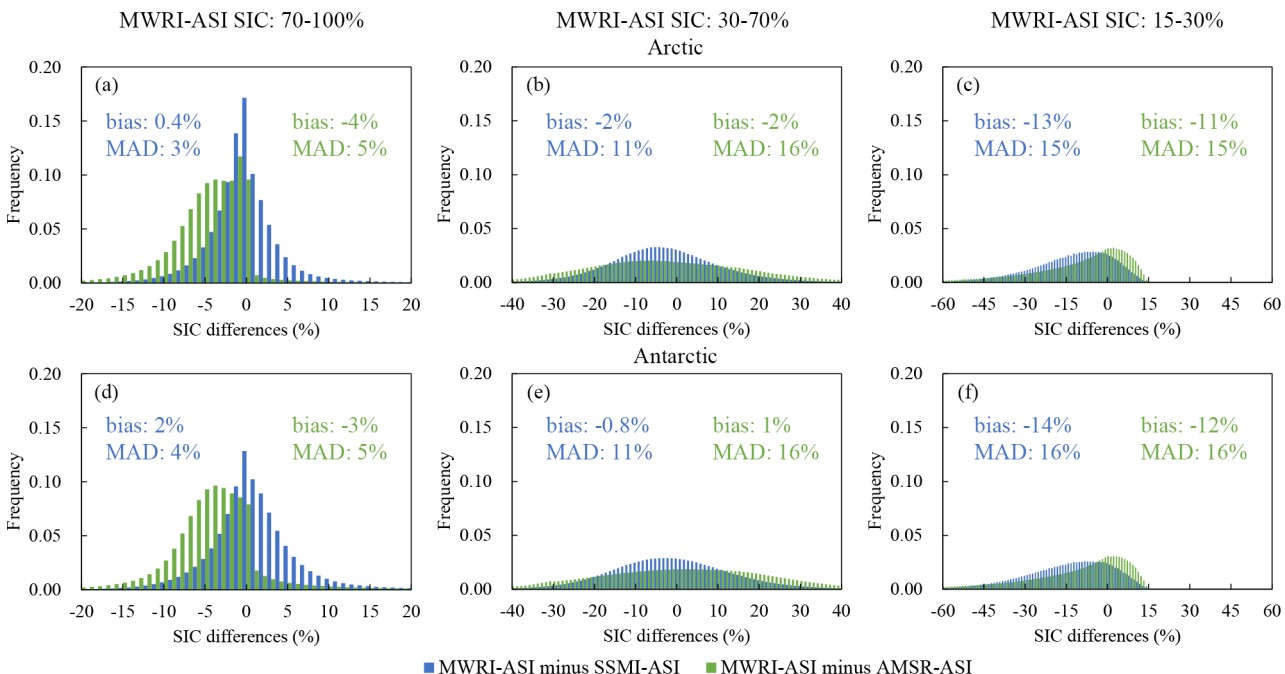

**Figure 4: Histogram of SIC differences between the MWRI-ASI and SSMI-ASI and between the MWRI-ASI and AMSR-ASI with the MWRI-ASI SIC of 70–100%, 30–70%, and 15–30% from November 2010 to December 2019. Also shown are the differences in SIC between the MWRI-ASI and SSMI-ASI (blue number) and between the MWRI-ASI and AMSR-ASI (green number).**

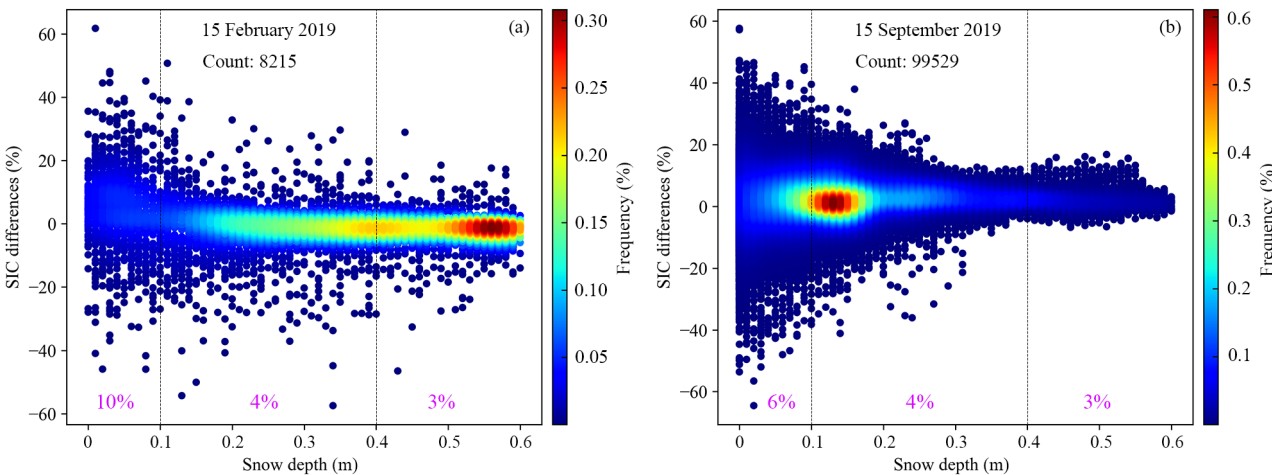

**Figure 5: Frequency of SIC differences between the MWRI-ASI and SSMI-ASI with different snow depths in the Antarctic on 15 February (a) and 15 September (b) 2019. The purple numbers present the MADs in SIC between the MWRI-ASI and SSMI-ASI with the snow depth of 0-0.1 m, 0.1-0.4 m, and 0.4-0.6 m.**

When the snow depth is less than 0.1 m, the SIC differences are the largest with a mean MAD of 7% between the MWRI-ASI and SSMI-ASI and of 9% between the MWRI-ASI and AMSR-ASI, which are about three times of those (2% and 3%) when

the snow depth is higher than 0.4 m (Table S2 of the supplement file). One of the reasons is that the TB differences are also largest when the snow depth is less than 0.1 m, about twice as much as when the snow depth is higher than 0.1 m (Table S3 of the supplement file). The spatial distribution of SIC differences do not show the obvious variability with the snow depth (Fig. 3), because all the SIC products are retrieved by the ASI algorithm, which have the consistent sensitivity to snow depth. In the Antarctic (Fig. 5), when the snow depth is less than 0.1 m, the MAD between MWRI-ASI SIC and SSMI-ASI SIC is 10% on the example day in summer, which are larger than those in winter by 4%. This could be explained by metamorphoses in the properties of snow over sea ice during summer, such as increased wetness (even saturated with meltwater), increased snow density, increased snow grain size, the occurrence of diurnal melt–refreeze cycles on surface, and slush on surface, etc., which have an impact on TB (Ivanova et al., 2015; Kern et al., 2016, 2019). The increase of snow wetness usually leads to an increase in TB of about 10 - 60 K, while the increase of snow grain size, which would cause the geophysical properties of snow cover to be very close to the surface scattering layer of sea ice, typically leads to a decrease in TB of about 15 - 35 K, resulting in large uncertainty of SIC (Kern et al., 2016). With the increase in snow depth, e.g., > 0.4 m, the corresponding increased snow load may lead to a negative ice freeboard, especially for the thin ice in the Antarctic, resulting in the slush layer appearing between the snow cover and the ice layer (Li et al., 2023). However, such slush layer is often thin, and the surface covered with thick snow would generally keep dry. This mechanism can be used to explain why the deviation of SIC is always the smallest for thick snow cover in both winter or summer. Thus, the snow over sea ice could play a significant role on the SIC uncertainties, which is greater at lower snow depth, especially in summer.

## 3.2 Comparison of the SIE products

In the Arctic, the MWRI-ASI SIE is smaller than the four existing SIE products of SSMI-NT, SSMI-BST, OSI-SAF, and Sea Ice Index, and has the smallest difference against the Sea Ice Index SIE with an overall MAD of $0.3 \times 10^6$ km$^2$ (Table 3). In the Antarctic, the MWRI-ASI SIE is larger than the Sea Ice Index SIE and lower than the other three SIE products, and it has the smallest differences against the Sea Ice Index SIE and OSI-SAF SIE, with an overall MAD of $0.2 \times 10^6$ km$^2$.

**Table 3. Biases and MADs between the MWRI-ASI SIE and the four existing SIE products during the entire overlap periods, as well as the winter and summer months from December 2010 to December 2019. The underlined numbers represent the lowest biases and MADs.**

|  |  | Overall ($10^6$ km$^2$) | | Winter ($10^6$ km$^2$) | | Summer ($10^6$ km$^2$) | |
| --- | --- | --- | --- | --- | --- | --- | --- |
|  |  | bias | MAD | bias | MAD | bias | MAD |
| Arctic | SSMI-BST | -0.9 | 0.9 | -0.9 | 0.9 | -0.9 | 0.9 |
|  | SSMI-NT | -0.6 | 0.6 | -0.7 | 0.7 | -0.5 | 0.5 |
|  | OSI-SAF | -0.7 | 0.7 | -0.7 | 0.7 | -0.7 | 0.7 |
|  | Sea Ice Index | -0.3 | 0.3 | -0.5 | 0.5 | -0.2 | 0.2 |
| Antarctic | SSMI-BST | -0.5 | 0.5 | -0.3 | 0.3 | -0.6 | 0.6 |
|  | SSMI-NT | -0.2 | 0.3 | <0.1 | 0.2 | -0.4 | 0.4 |
|  | OSI-SAF | -0.2 | 0.2 | -0.1 | 0.1 | -0.3 | 0.3 |
|  | Sea Ice Index | 0.2 | 0.2 | 0.4 | 0.4 | 0.1 | 0.1 |

From 2010 to 2019, the MWRI-ASI SIE shows a significant decline trend of -49,214 $km^2$ $yr^{-1}$ ($P<0.01$) in the Arctic and has the smallest difference in trends (-509 $km^2$ $yr^{-1}$ or about 1%) against the SSMI-NT SIE (Table 4). In the Antarctic, the largest reduction is identified for the MWRI-ASI SIE (-191,993 $km^2$ $yr^{-1}$, $P<0.01$), and the difference in trends between the MWRI-ASI SIE and OSI-SAF SIE is the lowest (-8,928 $km^2$ $yr^{-1}$ or about 5%).

In the period from 1979 to 2019, the four existing SIE products show significant reductions (about -55,000 $km^2$ $yr^{-1}$, $P<0.01$) in the Arctic and increasing trends (about 7,500 $km^2$ $yr^{-1}$, $P<0.01$) in the Antarctic (Table 4). The decreasing trends ($P<0.01$) of the combined SIEs in the Arctic are larger (by 15% to 40%) than those of the original SIEs because the MWRI-ASI SIE is lower than the four existing SIEs from 2010 to 2019. For the combination of the Sea Ice Index SIE and MWRI-ASI SIE in the Arctic, the differences in trends between the original and combined SIEs are the smallest (-8,343 $km^2$ $yr^{-1}$ or about 15%) compared to other combinations. In the Antarctic, the relatively small increasing trend of the original SSMI-BST SIE is reversed by the SIE combination of the SSMI-BST and MWRI-ASI due to lower MWRI-ASI SIE. The trend combining the Sea Ice Index SIE and MWRI-ASI SIE ($P<0.01$) in the Antarctic is larger than the original Sea Ice Index SIE trend because the MWRI-ASI SIE is higher than the Sea Ice Index SIE from 2010 to 2019. The combined Antarctic SIE trend of the OSI-SAF and MWRI-ASI ($P<0.05$) has the lowest differences (-4,736 $km^2$ $yr^{-1}$ or about 50%) against the original OSI-SAF SIE trend, compared to other combined SIE trends. Seasonally, in the Arctic, the differences in trends between the original and combined SIEs are larger in winter than in summer, because the differences between the MWRI-ASI SIE and four existing SIEs are larger in winter than in summer when more coastal sea ice are identified by the MWRI-ASI reducing the absolute deviations. In the Antarctic, the summer trends of the original SIEs are insignificant, but the significant winter increasing trends ($P<0.05$) are identified by all original SIEs and combined SIEs, except for the combination of the SSMI-BST SIE and MWRI-ASI SIE. Thereby, using the MWRI-ASI SIE instead of the original SIEs to construct a new time series has a greater impact on identifying the changing trend in SIE of the Antarctic than that of the Arctic because the Antarctic SIE trend is not as obvious as that of the Arctic.

For the Arctic annual maximum SIEs from 2011 to 2019, the ranking provided by the MWRI-ASI is the same as those provided by the SSMI-BST and Sea Ice Index, and all the largest values were observed in March 2012 for the five SIEs (Fig. 6). For the Arctic annual minimum SIEs, all the five SIEs identify the smallest and second-smallest values in September 2012 and 2019, respectively. The five SIEs have different rankings for 2013 and 2014 when the two largest Arctic annual minimum SIEs in 2011–2019 appeared, and the MWRI-ASI identifies the largest value in 2013, as the SSMI-BST and SSMI-NT. The five SIEs provide the same ranking of the Antarctic annual maximum SIEs and identify the largest and smallest SIEs in September 2014 and 2017, respectively, indicating the sudden drop after 2014 and rise after 2017 of Antarctic SIE can be depicted by all the five SIEs. The MWRI-ASI SIE reveals the smallest Antarctic annual minimum SIEs in 2017, as the SSMI-BST, OSI-SAF, and Sea Ice Index, but the SSMI-NT identifies the smallest value ($2.4\times10^6$ $km^2$) in 2018, which is slightly smaller than those in 2017 by $0.005\times10^6$ $km^2$. The MWRI-ASI has the same ranking for the five largest Antarctic annual minimum SIEs as the SSMI-BST and SSMI-NT. Thereby, the MWRI-ASI SIE is reasonable for identifying the extreme cases of both the annual

maximum and minimum SIEs in the bipolar regions, only with small differences appearing in some individual years with relatively low year-to-year differences.

**Table 4. Trends of the MWRI-ASI SIE and the four existing SIE products, and differences in trends between the MWRI-ASI SIE and the four existing SIE products from December 2010 to December 2019. Original trends of the four existing SIE products, combined trends (January 1979 to November 2010: four existing SIE products; December 2010 to December 2019: MWRI-ASI), and differences in trends between the original and combined SIEs from January 1979 to December 2019. The underlined numbers represent the lowest differences. The statistical significance at 95% and 99% confidence levels are marked by \* and \*\*, respectively.**

| | | 2010 – 2019 (km² yr⁻¹) | | 1979 – 2019 (km² yr⁻¹) | | |
|---|---|---|---|---|---|---|
| | | Trend | Differences | Original trend | Combined trend | Differences |
| Arctic | MWRI-ASI | -49,214 ± 10,884** | - | - | - | - |
| | SSMI-BST | -58,279 ± 12,654** | 9,065 | -57,644 ± 1,511** | -80,653 ± 1,831** | -23,009 |
| | SSMI-NT | -48,705 ± 12,672** | -509 | -55,879 ± 1,504** | -70,205 ± 1,608** | -14,326 |
| | OSI-SAF | -50,705 ± 11,529** | 1,491 | -53,379 ± 1,395** | -70,653 ± 1,567** | -17,274 |
| | Sea Ice Index | -45,527 ± 12,229** | -3,687 | -56,330 ± 1,489** | -64,673 ± 1,440** | -8,343 |
| Antarctic | MWRI-ASI | -191,993 ± 25,790** | - | - | - | - |
| | SSMI-BST | -181,269 ± 25,942** | -10,724 | 6,057 ± 2,192** | -5,834 ± 2,400* | -11,891 |
| | SSMI-NT | -178,209 ± 26,145** | -13,784 | 6,995 ± 2,189** | 1,628 ± 2,305 | -5,367 |
| | OSI-SAF | -183,065 ± 25,547** | -8,928 | 9,567 ± 2,171** | 4,831 ± 2,253* | -4,736 |
| | Sea Ice Index | -176,685 ± 25,461** | -15,308 | 7,436 ± 2,138** | 13,433 ± 2,192** | 5,997 |

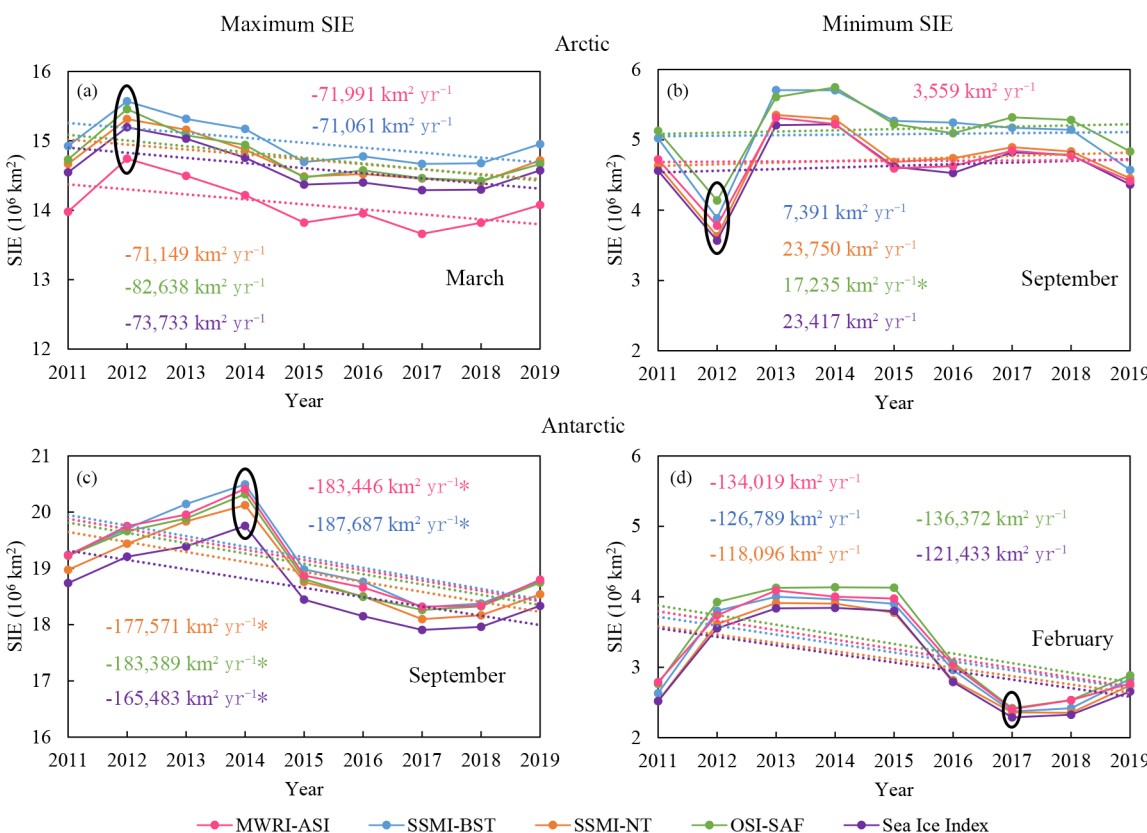

**Figure 6: Annual maximum and minimum SIEs from 2011 to 2019. Also shown are the corresponding SIE trends of the MWRI-ASI (pink), SSMI-BST (blue), SSMI-NT (orange), OSI-SAF (green), and Sea Ice Index (purple). The black circles present the largest annual maximum or the smallest annual minimum SIE. Note that the smallest Antarctic annual minimum SIE of the SSMI-NT is observed in February 2018.**

### 3.3 Comparison with ship-based SIC

The differences between the MWRI-ASI SIC and ship-based SIC are concentrated from -20% to 20%, generally accounting for 71% and 68% of the total samples in the Arctic and Antarctic, respectively (Fig. 7). In the region with high SIC (70–100%), about 82% of SIC differences between the MWRI-ASI and ship-based observation are distributed from -20% and 20%. However, this value decreases to about 67% and 46% in the regions with low SIC (15–30%) and medium SIC (30–70%), respectively. The MWRI-ASI SIC has smaller differences against the ship-based SIC in the regions with SIC above 70%, with MADs of 12% in both the Arctic and Antarctic, compared to those (16% and 17%) obtained for all samples.

Seasonally, in summer, the MADs between the MWRI-ASI SIC and ship-based SIC are 17% in the Arctic and 18% in the Antarctic, respectively, increasing by 5% compared to those in winter (Table 5). Higher accuracy of the MWRI-ASI SIC during winter compared to during summer is due to the high sensitivity of PM signal to atmospheric and ice surface melting conditions.

Spatially (Fig. 8), the SIC differences between the MWRI-ASI and ship-based observations within 50 km away from the coast are larger with MAD of 23% in the Arctic and of 22% in the Antarctic, respectively, compared to those (16%) beyond 50 km away from the coast. It illustrates that, compared to the ship-based observations, the MWRI-ASI SIC has comparable accuracy

with the SSMI-ASI SIC and AMSR-ASI SIC, although the accuracy of the MWRI-ASI SIC can be affected by the coast to a slight degree.

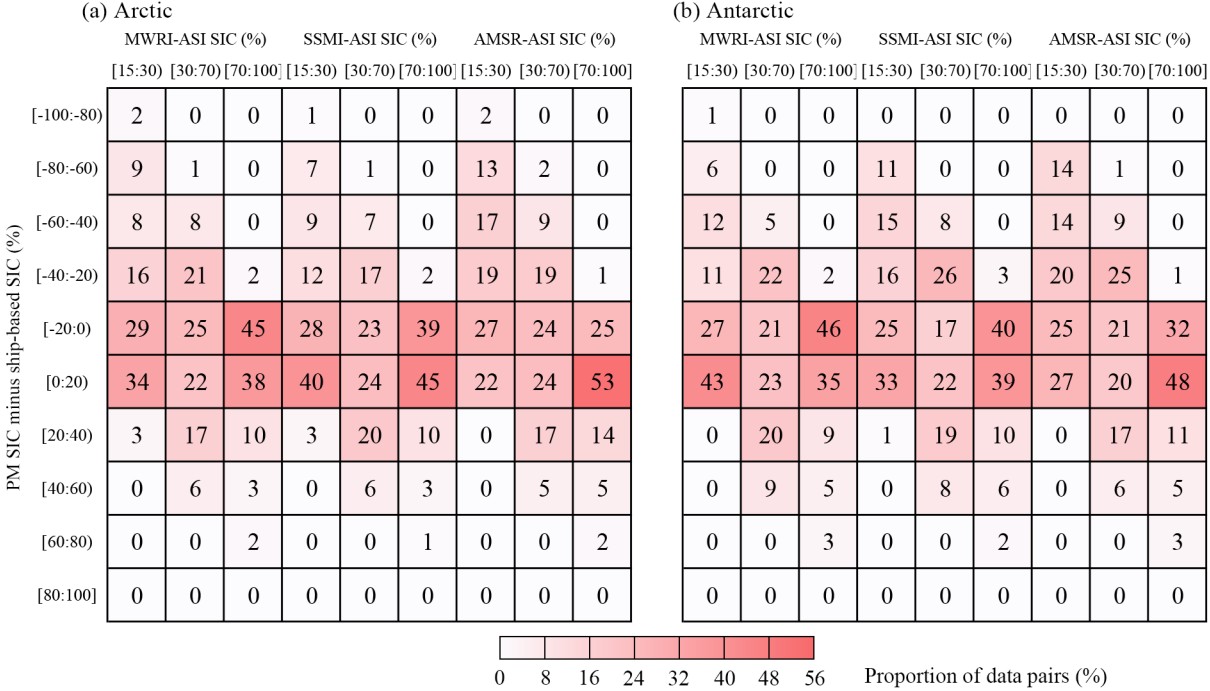

Figure 7: Proportion of data pairs (number in the grid) of the PM SIC products vs SIC differences between the PM SIC products and ship-based SIC. The PM SIC is divided to three categories of 15–30%, 30–70%, and 70–100% (horizontal axis). The SIC

differences are grouped with an interval of 20% (vertical axis).

**Table 5. Biases, MADs. RMSDs, and *Rs* between the PM SIC products and ship-based SIC during the entire overlap periods, as well as the summer and winter months from 2010 to 2019. The 'Count' is the number of data pairs of the individual PM SIC and ship-based SIC. The statistical significance at 95% and 99% confidence levels are marked by * and **, respectively.**

| | | Arctic | | | Antarctic | | |
|---|---|---|---|---|---|---|---|
| | | MWRI-ASI | SSMI-ASI | AMSR-ASI | MWRI-ASI | SSMI-ASI | AMSR-ASI |
| Overall | Count | 5230 | 5508 | 6169 | 2599 | 2613 | 2979 |
| | bias (%) | 2 | 3 | 6 | 3 | 1 | 4 |
| | MAD (%) | 16 | 15 | 16 | 17 | 18 | 16 |
| | RMSD (%) | 23 | 22 | 23 | 23 | 24 | 23 |
| | *R* | 0.60** | 0.66** | 0.57** | 0.62** | 0.57** | 0.57** |
| Winter | Count | 649 | 655 | 670 | 696 | 794 | 910 |
| | bias (%) | -4 | -4 | -0.2 | 3 | 0.7 | 3 |
| | MAD (%) | 12 | 11 | 7 | 14 | 14 | 13 |
| | RMSD (%) | 19 | 18 | 15 | 19 | 20 | 20 |
| | *R* | 0.44** | 0.53** | 0.42** | 0.67** | 0.64** | 0.58** |
| Summer | Count | 4581 | 4853 | 5499 | 1903 | 1819 | 2069 |
| | bias (%) | 2 | 3 | 7 | 3 | 2 | 5 |
| | MAD (%) | 17 | 16 | 17 | 18 | 20 | 18 |
| | RMSD (%) | 23 | 22 | 24 | 24 | 26 | 25 |
| | *R* | 0.59** | 0.66** | 0.55** | 0.59** | 0.53** | 0.56** |

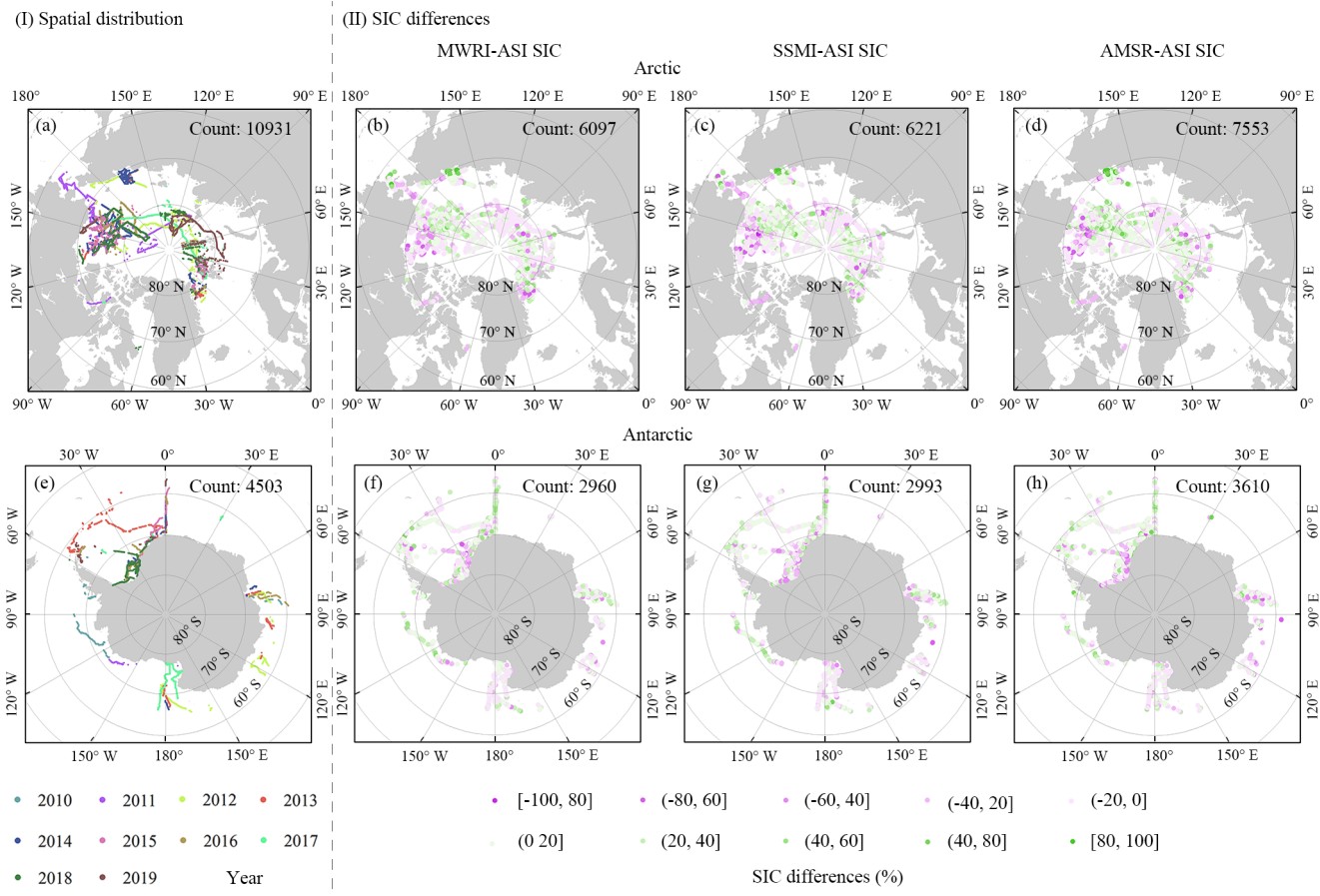

**Figure 8: (I). Spatial distributions of the ship-based observational SIC in 2010–2019. (II). SIC differences between the PM SIC products and ship-based observations.**

## 4 Discussion

### 4.1 TB differences among the MWRI, SSMI, and AMSR series sensors

At the frequencies used for calculating SIC, i.e., at 89 GHz with V- and H-polarization and differences between them (defined as polarization differences), the MWRI TB is closer to the AMSR TB than the SSMI TB (Table 6). In the PIZ, the MADs of polarization difference at 89 GHz between the MWRI TB and SSMI TB are smaller with values of 1.5 and 1.7 K in the Arctic and Antarctic, respectively, compared to those in the MIZ (4.6 and 4.4 K) and over open water (7.6 and 7.6 K), where the atmospheric influence is stronger. Overall, the polarization difference at 89 GHz of the MWRI sensor is slightly higher than that of the SSMI sensor with a mean positive bias of 0.9 K and slightly lower than that of the AMSR sensor with a mean negative bias of -1.1 K. It illustrates that the re-calibrated MWRI TB is comparable to the SSMI and AMSR TB, which can be well applied to the ASI algorithm.

The TB differences between the MWRI and SSMI sensor are smaller than those between MWRI and AMSR sensor at low frequencies used for the weather filters. The overall biases between MWRI TB and SSMI TB are -1.2, -0.7, and 0.1 K at 18.7, 23.8, and 36.5 GHz with V-polarization, respectively. Compared to the AMSR TB, the MWRI TB is lower with mean biases of -5.3, -5.4, and -3.9 K at 18.7, 23.8, and 36.5 GHz with V-polarization, respectively. In general, the low frequencies of MWRI sensor can filter the weather effects, which is similar to those of the SSMI and AMSR sensors.

**Table 6. MADs between the MWRI TB and other two TBs in the PIZ, MIZ, and open water from 2010 to 2019. The first and second numbers in each cell present the MADs between the MWRI TB and SSMI TB, and those between the MWRI TB and AMSR TB.**

|  | Arctic (K) | | | Antarctic (K) | | |
|---|---|---|---|---|---|---|
|  | PIZ | MIZ | Open water | PIZ | MIZ | Open water |
| 18 V | 1.7 / 4.0 | 4.8 / 7.2 | 7.4 / 11.4 | 2.4 / 4.5 | 3.6 / 6.6 | 4.8 / 9.5 |
| 23 V | 2.0 / 4.1 | 4.5 / 6.2 | 7.0 / 11.4 | 2.3 / 4.7 | 3.5 / 6.5 | 4.3 / 9.1 |
| 36 V | 2.5 / 2.8 | 4.1 / 4.7 | 6.3 / 9.2 | 2.5 / 3.0 | 3.4 / 4.9 | 4.2 / 6.3 |
| 89 V | 3.9 / 3.5 | 4.6 / 3.4 | 13.8 / 7.1 | 4.1 / 3.3 | 4.2 / 3.6 | 8.1 / 4.0 |
| 89 H | 4.3 / 3.8 | 7.4 / 5.5 | 18.1 / 10.7 | 4.9 / 3.8 | 7.1 / 5.4 | 14.1 / 8.3 |
| 89 V – 89 H | 1.5 / 1.3 | 4.6 / 4.1 | 7.6 / 6.0 | 1.7 / 1.5 | 4.4 / 3.8 | 7.6 / 5.7 |

### 4.2 Improvements and limitations of the MWRI-ASI v2 SIC

Compared to the previous version of MWRI-ASI SIC generated by Zhao et al. (2022) (MWRI-ASI v1 SIC), the MWRI-ASI SIC generated by this study (MWRI-ASI v2 SIC) is lower by -3% in the Arctic in 2018, especially in the MIZ and in summer. The MWRI-ASI v1 SIC has small differences against AMSR-ASI SIC, but the time series of AMSR-ASI is shorter than the SSMI-ASI SIC. To obtain longer-term SIC products, this study modified the previous algorithm and adopted the SSMI-ASI SIC as referential SIC to update and extend the MWRI-ASI SIC. The MWRI TB applied for the MWRI-ASI v1 SIC were bias-corrected using the AMSR2 TB by a linear aggression during only one-year overlap period of 2018. As a result, the systematic deviations between the MWRI-ASI v1 SIC and AMSR-ASI SIC increased in 2019 compared to those in 2018, especially in summer (Chen et al., 2021). To obtain a more consistent SIC product, the re-calibrated MWRI TB were adopted as input source TB, because the differences among the re-calibrated MWRI TB from different satellites were low with mean of 2.8 and 1.0 K in the Arctic and Antarctic, respectively. From 2018 to 2019, when only the FY-3D MWRI TB were used, the differences in systematic deviations between the MWRI-ASI v2 SIC and AMSR-ASI SIC were 0.6% in summer, which were relatively small compared to those (1%) between the MWRI-ASI v1 SIC and AMSR-ASI SIC. It indicates that the consistency of the MWRI-ASI v2 SIC is improved compared with the MWRI-ASI v1 SIC.

However, the MWRI-ASI SIC v2 is still limited in terms of the land spillover and weather effects. Due to the differences in the size of the view field of different TB frequencies and the differences in observational TB of open water and land, the spurious sea ice would emerge in the PM observations along the coasts (Lavergne et al., 2019; Kern et al., 2019). Although some methods have been proposed to solve the land spillover by expanding the land mask (Maslanik et al., 1996), subtracting

the summer minimum SIC from original images (Cavalieri et al., 1996), and estimating the fraction of land emissivity in the TB (Maaß and Kaleschke, 2010), the SIC differences are still higher in the near-coast regions than in the regions far away

from the coast. The SIC MADs between the MWRI-ASI and SSMI-ASI within 50 km away from the coast are larger with values of 6% in the Arctic and of 7% in the Antarctic, compared to those (4% and 5%) beyond 50 km away from the coast. The MADs within 50 km away from the coast between MWRI-ASI SIC and AMSR-ASI SIC are 8% in the Arctic and 9% in the Antarctic, which are higher than those (5% and 6%) beyond 50 km away from the coast. Compared to the SSMI-ASI and AMSR-ASI, the MWRI-ASI reveals more ice along the coasts and around the islands, such as around 72° N from 138° to 144°

E in the Arctic and around 78° S from 160° W to 170° E in the Antarctic (Fig. 9). The ice along the coast extends about two grids (25 km) from the coastline, leading to the overestimation of the MWRI-ASI SIE in summer compared to the SSMI-ASI SIE. Due to larger uncertainties of our MWRI-ASI SIC in the near-coast region, it is recommended that the grids extended outward from the coast by 50 km can be removed when using it.

To analyze the influence of temporal filter on land spillover, we acquired the single-day SSMI-ASI SIC product (Single-day

SSMI-ASI) from the French Research Institute for Exploitation of the Sea via the Centre d'Exploitation et de Recherche SATellitaire (Ifremer/CERSAT) (Girard-Ardhuin et al., 2008). The SSMI-ASI SIC produced by Hamburg Uni. were filtered by a five-day median filter (Five-day SSMI-ASI). In the region within 50 km away from the coast, the SIC MADs between MWRI-ASI and Single-day SSMI-ASI are 6% in both the Arctic and the Antarctic, which are slightly smaller than those between MWRI-ASI and Five-day SSMI-ASI by 0.4%. It indicates that the SIC uncertainties in the near-coast regions would

be slightly increased after temporal filter.

The spurious sea ice over open water arises from the atmospheric effect, i.e., water vapor, cloud liquid water, surface winds, and precipitation (Kern, 2004). Although the methods to remove the spurious ice have been applied, some spurious floes remain and some small real floes are ignored in the MIZ. The MWRI-ASI weather filters can remove most of the spurious ice (Fig, 9). However, after applying the weather filters, more ice floes have been identified by the MWRI-ASI in the MIZ

compared to the SSMI-ASI and AMSR-ASI. It is still difficult to determine whether this residual ice is erroneous ice caused by weather effects or the real discrete small ice floes. Thus, in the future work, we will attempt to identify and remove the spurious ice caused by land spillover and weather effects, by combining the optical or synthetic aperture radar images with higher resolutions, to further improve our MWRI-ASI SIC product.

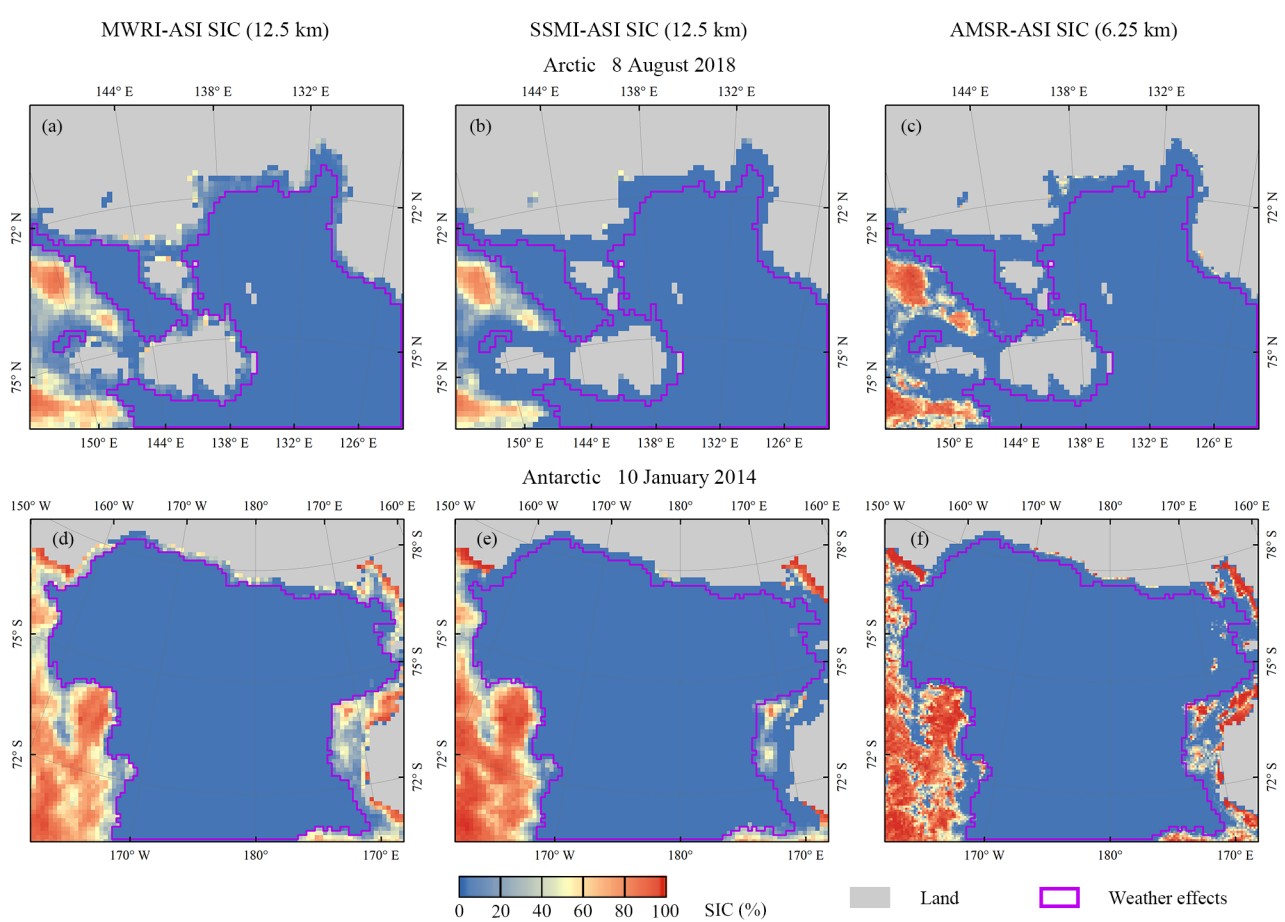

**Figure 9: SIC distributions of the MWRI-ASI, SSMI-ASI, and AMSR-ASI in the coastal regions. Also shown are the weather-affected grids removed by the MWRI-ASI weather filters.**

## 4.3 Comparisons to prior studies on SIC products in the polar regions

The biases between our MWRI-ASI SIE and Sea Ice Index SIE from 2015 to 2017 are -0.3×10$^6$ km$^2$ in the Arctic and 0.2×10$^6$ km$^2$ in the Antarctic, respectively, which are within the range of SIE biases in the same period shown in Meier and Stewart

(2019). Seasonally, in the bipolar regions, our MWRI-ASI SIE has larger absolute deviations against the Sea Ice Index SIE in winter than in summer. The reduction of absolute deviation of Arctic SIE in summer is because our MWRI-ASI SIE has been involved more sea ice along the coastline and at the ice edge in summer than in winter. In the Antarctic, our MWRI-ASI SIE is about two grid cells (25 km) farther south than the Sea Ice Index SIE, and the absolute deviation increases in winter because the SIE is larger in winter than in summer. The seasonal variation pattern of biases between our MWRI-ASI SIE and Sea Ice

Index SIE is similar to those between the AMSR-BST SIE and Sea Ice Index SIE in the Arctic (Meier and Stewart, 2019). Beitsch et al. (2015) revealed that the accuracies of the AMSR-E ASI SIC and SSMI-ASI SIC are consistent, with overall RMSDs of about 13% compared to the ASPeCt observations from 2002 to 2010. Our study also indicates that the accuracy of

the MWRI-ASI SIC is comparable to those of the SSMI-ASI SIC and AMSR-ASI SIC, with overall RMSDs of about 23% compared to the ship-based observations from 2010 to 2019. The accuracy of our MWRI-ASI SIC is lower in summer than in winter, which is consistent with those of the AMSR-E ASI SIC and SSMI-ASI SIC (Beitsch et al., 2015). This study and the results given by Spreen et al., (2008) both presented that the SIC differences between PM-based and ship-based observations are larger in the low-SIC region than those in the high-SIC region. The large SIC differences can be explained by the different spatial and temporal scales between PM SICs and ship-based SICs (Beitsch et al., 2015; Kern et al., 2019). Ship-based SICs are obtained on an elliptically shaped area of 1 km on each side of the ship, while the footprint sizes of PM frequencies were considerably larger than 1 km, which are several kilometers to tens of kilometers. In contrast to ship-based SIC gained by observers at a specific time, the PM SICs are the daily averages combined with swath SICs from different time in one calendar day. The ship-based SIC may not be fully representative of the entire grid of PM SIC and the observation results may also be affected by visibility and light around the ship.

## 5 Data availability

The SIC product is derived from the FY-3 MWRI sensors, which can be downloaded from the data repositories PANGAEA at https://doi.pangaea.de/10.1594/PANGAEA.945188 (Chen et al., 2022). This dataset is available from 12 November 2010 to 31 December 2019 with temporal data gaps of 23 days in the Arctic and 82 days in the Antarctic. The SIC files are named "FY_MWRI_SIC_DAILY_YYYYMMDD_Region.tif", with "YYYYMMDD" denoting the date and "Region" representing the Arctic or Antarctic. This SIC dataset is archived in TIFF format and can be read using Python, ENVI/IDL, and MATLAB software. The values '0-100' are the percentage of SIC, flag of '-1' is the land and of '-2' is the Pole Hole, and 'NoData' for missing data. Additionally, the biases between this SIC dataset and other two ASI SIC products, i.e., SSMI-ASI and AMSR-ASI, are provided.

## 6 Conclusion

This study generates a new SIC product in the polar regions from November 2010 to December 2019, which is derived from the recent re-calibrated TB data of the MWRI sensors onboard the FY-3B, FY-3C, and FY-3D satellites using the modified dynamic tie points ASI algorithm. Generally, the MWRI-ASI SIC or SIE can reasonably identify the seasonal and long-term changes in sea ice, as well as the extreme cases of annual maximum/minimum SIE for both the Arctic and Antarctic.

To test the skill of the MWRI-ASI as an independent PM SIC dataset, the MWRI-ASI SIC is compared to the existing ASI SIC products of SSMI-ASI and AMSR-ASI, and the MWRI-ASI SIE is compared to the existing SIE products of SSMI-BST, SSMI-NT, OSI-SAF, and Sea Ice Index. The accuracy of the MWRI-ASI SIC is also validated using the ship-based observed SIC.

Both the daily SIC and SIE derived from the MWRI-ASI closely agree with those derived from the SSMI-ASI, with overall SIC biases of -1% and 0.5%, and with overall SIE biases of $0.1\times10^6$ km$^2$ and $-0.02\times10^6$ km$^2$ in the Arctic and Antarctic, respectively. The ability of the MWRI-ASI to identify the MIZ mostly coincides with that of the SSMI-ASI. Therefore, the

MWRI-ASI SIC is closer to the SSMI-ASI SIC compared to AMSR-ASI SIC. The MWRI-ASI SIC has larger uncertainties in the region with low or medium SIC than in the region with high SIC. Shallower snow depth over sea ice cause larger uncertainties of SIC, especially during the summer. The sensitivity of MWRI-ASI SIC to sea ice melting surface is higher than the SSMI-ASI, which suggests the MWRI-ASI SIC product may have better ability to identify the melting surface.

The MWRI-ASI SIE has smaller differences against the Sea Ice Index SIE in the Arctic and against the OSI-SAF SIE in the

Antarctic compared to other products. Based on the comparison with the ship-based observations, the accuracy of the MWRI-ASI SIC is comparable to those of the SSMI-ASI SIC and AMSR-ASI SIC. It suggests the MWRI-ASI SIC product can be independently used for monitoring changes in sea ice and serve as reliable data to evaluate the next-generation sensors.

Future improvements are aimed to identify and remove the spurious sea ice caused by land spillover and weather effects more accurately by using satellite-based observations with higher resolutions to further improve the MWRI-ASI SIC. In addition,

based on the re-calibrated TB data of the MWRI sensors, we can produce other geophysical variables for the sea ice in the polar regions, e.g., lead fraction, onsets of ice surface melt or freeze, and other morphological characteristics.

## Appendix A: Abbreviations

**Table A1. List of abbreviations used in the paper.**

| Abbreviation | Term |
| --- | --- |
| SIC | Sea ice concentration |
| SIE | Sea ice extent |
| PM | Passive microwave |
| MWRI | Microwave Radiation Imager |
| FY-3 | FengYun-3 |
| SSMI | Special Sensor Microwave Imager series |
| SMMR | Scanning Multichannel Microwave Radiometer |
| SSM/I | Special Sensor Microwave/Imager |
| SSMIS | Special Sensor Microwave Imager Sounder |
| AMSR | Advanced Microwave Scanning Radiometer series |
| AMSR2 | Advanced Microwave Scanning Radiometer 2 |
| AMSR-E | Advanced Microwave Scanning Radiometer-EOS |
| MIZ | Marginal ice zone |
| PIZ | Pack ice zone |
| ASI | Arctic Radiation and Turbulence Interaction Study Sea Ice |
| BST | Bootstrap |
| NT2 | Enhanced NASA Team |

| | |
|---|---|
| NT | NASA Team |
| PMA | Passive microwave algorithm |
| NSMC | Chinese National Satellite Meteorological Center |
| OSI-SAF | Ocean and Sea Ice Satellite Application Facility |
| NSIDC | National Snow and Ice Data Center |
| ICDC | Integrated Climate Data Center |
| GESR | Goddard Earth Science Research |
| TB | Brightness temperature |
| H | Horizontal polarization |
| V | Vertical polarization |
| GR | Gradient ratio |
| Ice Watch/ASSIST | Ice Watch/Arctic Ship-based Sea-Ice Standardization |
| ASPeCt | Antarctic Sea Ice Processes and Climate |
| CHINARE | Chinese National Arctic and Antarctic Research Expedition |
| MAD | Mean absolute deviation |
| RMSD | Root mean standard deviation |
| $R$ | Correlation coefficient |

**Author Contributions.**

YC performed the experiments and wrote the manuscript. XP, RL, and XZ provided the conception of the study and suggestions on manuscript. SW and PZ provided valuable instructions on data and methods. YL, PF, and QJ advised on the model code. All authors contributed to the improvement of the manuscript.

**Competing interests.**

The authors declare that they have no conflict of interest.

**Acknowledgements.**

We would like to acknowledge the organizations that shared their datasets for use in this study. The Chinese National Satellite Meteorological Center provided the re-calibrated MWRI TB dataset. The Integrated Climate Data Center (ICDC) of the University of Hamburg provided the SSMI-ASI SIC product and ESA-CCI ship-based observed SIC. The Institute of Environmental Physics of the University of Bremen provided the AMSR2-ASI SIC product. The U.S. Geological Survey 515 (USGS), NASA National Snow and Ice Data Center provided the TB dataset (SSMI TB and AMSR TB), snow depth data, and SIE dataset (SSMI-BST, SSMI-NT, and Sea Ice Index SIE). The Exploitation of Meteorological Satellites (EUMETSAT) Ocean and Sea Ice Satellite Application Facility (OSI-SAF) provided the OSI-SAF SIE datasets. The U.S. Geological Survey (USGS), NASA Goddard Earth Science Projects provided the Arctic sea ice surface melt/freeze onset data. The French

Research Institute for Exploitation of the Sea provided the Single-day SSMI-ASI SIC product. We also want to acknowledge the contributors of the ship-based observed SIC in the IceWatch and PANGAEA databases, as well as during the CHINARE cruises. In addition, we are grateful to the reviewers and editors for their improvements to this paper.

**Financial support.**

This work was supported by the National Key Research and Development Program of China (grant Nos. 2018YFA0605903, 2021YFC2801304, and 2018YFB0504905), the National Natural Science Foundation of China (grant Nos. 42076235 and 41876223), the Fundamental Research Funds for the Central Universities (grant No. 2042022kf0018), and the Program of Shanghai Academic/Technology Research Leader (grant No. 22XD1403600).

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
