# Peer review of "A new sea ice concentration product in the polar regions derived from the FengYun-3 MWRI sensors"

_Earth System Science Data, 2022_

## Author Comment (AC1)

*I am pleased to see that the ASI method is being used successfully on the Chinese satellites FengYun-3B, 3C and 3D. To my knowledge this is the first sea ice product from a Chinese satellite series that covers a long time period. Therefore, I consider this work potentially relevant for the ESSD journal. I'd like to give only general comments and questions because the information provided with the manuscript does not allow further review in particular with respect to the source data and validation.*

*A code repository is missing, see https://essd.copernicus.org/articles/10/2275/2018/*

**Reply:**

Thanks for your comment. We have provided the processing steps of our MWRI-ASI SIC product in the supplement file, including the flow chart, steps, codes, and procedure outcomes.

*For the introduction of the near 90 GHz method please also refer to the prior work of Svendsen et al. (1987) which forms the basics of the algorithm. Svendsen, E., Matzler c., and Grenfell, TC. (1987). "A Model for Retrieving Total Sea Ice Concentration from a Spaceborne Dual-Polarized Passive Microwave Instrument Operating Near 90 GHz", International Journal 0/ Remote Sensing, Vol. 8, No. 10, pp. 1479-1487.*

**Reply:**

Thanks for your advice. We have added this reference.

Page 6, Line 158-159, we wrote:

More details about the dynamic tie points ASI algorithm were given in Zhao et al. (2022), and the details about the ASI algorithm can be referred to Svendsen et al., (1987), Kaleschke et al. (2001) and Spreen et al. (2008).

*I can not judge the quality of the brightness temperature because it is not yet published: The recently re-calibrated brightness temperature (TB) of the MWRI sensors provided by NSMC (Wu et al., 2022) were used in this study. Because ESSD is a journal for "open data" (isn't it?), I also would like to know a bit more about the availability of the source data. Table 1: TB characteristics missing. What about uncertainties? Grid resolution is different from field of view. Table 1 in Zhao et al. (2021) states across scan resolution of 89 GHz is 9x15 km^2. With a grid spacing of 12.5 km there is significant undersampling in one direction. Why not use a 6 km grid for the sake of Nyquist-Shannon?*

**Reply:**

Thanks for your comments and questions. We have removed the reference "Wu et al., 2022", which is still under review. Once it is published, we will add it again. We added the website for the re-calibrated MWRI TB data. We also added the inter-comparison among the MWRI TB, SSMI TB, and AMSR TB in Section 4.1 to illustrate the quality and uncertainties of the re-calibrated MWRI TB. The grid resolution is determined based on time integration, and the distance between two points is 10 km, which is closer to the 12.5-km grid than the 6.25-km grid. Therefore, we projected the re-calibrated MWRI TB on the 12.5-km polar stereographic grid.

Page 3, Line 91-93, we wrote:

Therefore, the MWRI TB data was re-calibrated using the operational algorithm, which focused on the hot load, antenna, and receiver calibration, reducing the TB deviations of different MWRI sensors.

Page 3, Line 94-95, we wrote:

This study used the re-calibrated level 1 swath MWRI TB data from the FY-3B, FY-3C, and FY-3D satellites, provided by the NSMC, which is available at http://www.richceos.cn (Table 1).

Page 4, Line 103-110, we wrote:

To evaluate the uncertainties of the re-calibrated MWRI TB in the polar regions, we chose two daily TB products, i.e., the SSMI TB (version 6, Meier et al., 2021) and AMSR TB (AMSR-E version 3, Cavalieri et al., 2014; AMSR2 version 1, Meier et al., 2018), which are both available from the National Snow and Ice Data Center (NSIDC). This SSMI TB product is projected on 12.5-km and 25-km polar stereographic grids at high and low frequencies, respectively. All the frequencies of the AMSR TB products are projected on a 12.5-km polar stereographic grid. The time coverages of the two daily TB products are corresponding to that of the MWRI TB. To conduct a comparison among these three TB products, the swath MWRI TB was gridded to daily MWRI TB and the low frequencies of SSMI TB were resampled to the 12.5-km polar stereographic grid. The TB differences were calculated in the PIZ, MIZ, and open water.

Page 18-19, Line 371-383, we wrote:

For the frequencies used for the ASI algorithm, i.e., at 89 GHz with V- and H-polarization and differences between them (defined as polarization differences), the MWRI TB is closer to the AMSR TB than the SSMI TB (Table 6). In the PIZ, the MADs of polarization difference at 89 GHz between the MWRI TB and SSMI TB are smaller with values of 1.5 and 1.7 K in the Arctic and Antarctic, respectively, compared to those (4.6 and 4.4 K; 7.6 and 7.6 K) in the MIZ and over open water, where the atmospheric influence is more intense. Overall, the polarization difference at 89 GHz of the MWRI sensor is lightly higher than that of the SSMI sensor with a mean positive bias of 0.9 K and lightly lower than that of the AMSR sensor with a mean negative bias of -1.1 K. It illustrates that the re-calibrated MWRI TB is comparable to the SSMI and AMSR TB, which can be well applied to the ASI algorithm.

The TB differences between the MWRI and SSMI sensor are smaller than those between MWRI and AMSR sensor at low frequencies used for the weather filters. The overall biases between MWRI TB and SSMI TB are -1.2, -0.7, and 0.1 K at 18.7, 23.8, and 36.5 GHz with V-polarization, respectively. Compared to the AMSR TB, the MWRI TB is lower with mean biases of -5.3, -5.4, and -3.9 K at 18.7, 23.8, and 36.5 GHz with V-polarization, respectively. In general, the low frequencies of MWRI sensor can filter the weather effects, which is similar to those of the SSMI and AMSR sensors.

**Table 6. MADs between the MWRI TB and other two TBs from 2010 to 2019 in the PIZ, MIZ, and open water. "A / B": A present the MADs between the MWRI TB and SSMI TB, and B present the MADs between the MWRI TB and AMSR TB.**

|  | Arctic | | | Antarctic | | |
| --- | --- | --- | --- | --- | --- | --- |
|  | PIZ | MIZ | Open water | PIZ | MIZ | Open water |
| 18 V | 1.7 / 4.0 | 4.8 / 7.2 | 7.4 / 11.4 | 2.4 / 4.5 | 3.6 / 6.6 | 4.8 / 9.5 |
| 23 V | 2.0 / 4.1 | 4.5 / 6.2 | 7.0 / 11.4 | 2.3 / 4.7 | 3.5 / 6.5 | 4.3 / 9.1 |
| 36 V | 2.5 / 2.8 | 4.1 / 4.7 | 6.3 / 9.2 | 2.5 / 3.0 | 3.4 / 4.9 | 4.2 / 6.3 |
| 89 V | 3.9 / 3.5 | 4.6 / 3.4 | 13.8 / 7.1 | 4.1 / 3.3 | 4.2 / 3.6 | 8.1 / 4.0 |
| 89 H | 4.3 / 3.8 | 7.4 / 5.5 | 18.1 / 10.7 | 4.9 / 3.8 | 7.1 / 5.4 | 14.1 / 8.3 |
| 89 V – 89 H | 1.5 / 1.3 | 4.6 / 4.1 | 7.6 / 6.0 | 1.7 / 1.5 | 4.4 / 3.8 | 7.6 / 5.7 |

Page 19, Line 394-396, we wrote:

To obtain a more consistent SIC product, the re-calibrated MWRI TB were adopted as input source TB, because the differences among the re-calibrated MWRI TB from different satellites were low with mean of 2.8 and 1.0 K in the Arctic and Antarctic, respectively.

*I have not fully understood the concept of the preliminary dynamic tie points. This could be outlined in more detail.*

**Reply:**

Thanks for your suggestion. The "preliminary dynamic tie points" presented in the Abstract is the ASI algorithm in Zhao et al. (2022): Zhao, X., Chen, Y., Kern, S., Qu, M., Ji, Q., Fan, P., and Liu, Y.: Sea Ice Concentration Derived from FY-3D MWRI

and Its Accuracy Assessment, IEEE Trans. Geosci. Remote Sens., 60, https://doi.org/10.1109/TGRS.2021.3063272, 2022, which is our previous study. In this study, we modified the ASI algorithm of Zhao et al. (2022), presented in Section 2.4. To better understand this sentence, we rewrote it. Meanwhile, we have showed procedures of the initial tie points and dynamic tie points in more detail in the supplement file (S1.2 and S1.4).

Page 1, Line 17-18, we wrote:

We modified the previous Arctic Radiation and Turbulence Interaction Study Sea Ice (ASI) dynamic tie points algorithm mainly by changing input brightness temperature and initial tie points.

Page 6, Line 161-162, we wrote:

The detailed procedures for retrieving this MWRI-ASI SIC product can be seen in the supplement file.

Page 7, Line 176-177, we wrote:

The steps for generating the initial points are given in more details in Section S1.2 of the supplement file.

Page 7, Line 182-183, we wrote:

The Section S1.4 of the supplement file illustrates detailed procedures of generating the dynamic tie points.

Page 19, Line 389-391, we wrote:

The MWRI-ASI v1 SIC has small differences against AMSR-ASI SIC, but the time series of AMSR-ASI is shorter than the SSMI-ASI SIC. To obtain longer-term SIC products, this study modified the previous algorithm and adopted the SSMI-ASI SIC as referential SIC to update and extend the MWRI-ASI SIC.

*My main concern is insufficient scientific motivation: "In order to promote the application of MWRI sensors, especially to back up the existing sea ice products". Why is there the need for a backup for the existing data? Well, I could understand the argument for the future. This could be further elaborated. However, I think the main advantage of having such a data set is the true independence which potentially allows the application of techniques like triple collocation between different satellite data records. How would you judge the "risk of breaking". For example, we expect the AMSR3 launch in 2023 or 24.*

**Reply:**

Thanks for your suggestion. We have reconsidered the motivation of our studies and pointed out that our MWRI-ASI SIC product can be as independent SIC dataset to monitor the changes of sea ice and serve as reliable data to evaluate the future sensors. Meanwhile, the "risk of breaking" was not highlighted, and we rewrote the motivation.

Page 2, Line 35-49, we wrote:

Due to the long-term services of spaceborne sensors, the PM sea ice measurements can continuously track the response of sea ice to climate change and support the applications for climate models or multidisciplinary studies in the polar regions. The currently operating Special Sensor Microwave Imager Sounder (SSMIS) and Advanced Microwave Scanning Radiometer 2 (AMSR2) sensors have been used for many years beyond their design lifetime (Gerland et al., 2019). The new missions for successors of these two sensors or the launch plans for other instruments, e.g., the Copernicus Imaging Microwave Radiometer (Jiménez et al., 2021) and Weather Satellite Follow–On–Microwave (Newell et al., 2020), are in the preparation stage and will be achieved in the next several years. The Chinese instruments, the Microwave Radiation Imager (MWRI) sensors onboard the FengYun-3 (FY-3) series satellites, i.e., FY-3A, FY-3B, FY-3C, and FY-3D (Zhang et al., 2018, 2019; Xian et al., 2021), are promisingly used to independently provide long-term SIC (Chen et al., 2021; Zhao et al., 2022). However, the inconsistency of different sensors and the drift of the sensor itself with increased operation time can increase the uncertainties of SIC (Eisenman et al., 2014). Thus, enriching data resources are beneficial to achieve the triple collocation between different

satellites. For example, the SSMIS data has been used as a bridge to compare and connect the Advanced Microwave Scanning Radiometer-EOS (AMSR-E) and AMSR2 estimates, which have a data gap from 2011 to 2012 (Meier and Ivanoff, 2017). Besides, a new SIC product from the MWRI sensors after systematic assessment can also provide an option to verify the SIC products of new launched PM sensors in the future.

Page 3, Line 78-79, we wrote:

In order to promote the application of MWRI sensors, this study extends the work of Zhao et al. (2022) and generates a new polar SIC product from November 2010 to December 2019.

Page 3, Line 81-84, we wrote:

Moreover, the MWRI-ASI SIC is compared to the existing ASI SIC products and assessed systematically using ship-based SIC observation to identify its uncertainty in various regions and seasons. We also derive SIE from the MWRI-ASI SIC and compare it to the existing SIE products to test its ability to independently monitor sea ice changes.

Page 5-6, Line 149-150, we wrote:

Moreover, to test the capability of the MWRI-ASI SIE as an independent indicator for climate changes, we performed an analysis of combined SIE trends.

Page 22, Line 462-465, we wrote:

To test the ability of the MWRI-ASI as an independent PM SIC dataset, the MWRI-ASI SIC is compared to the existing ASI SIC products of SSMI-ASI and AMSR-ASI, and the MWRI-ASI SIE is compared to the existing SIE products of SSMI-BST, SSMI-NT, OSI-SAF, and Sea Ice Index. The accuracy of the MWRI-ASI SIC is also validated using the ship-based observed SIC.

Page 23, Line 475-476, we wrote:

It suggests the MWRI-ASI SIC product can independently monitor changes of sea ice and serve as reliable data to evaluate the future sensors.

*What is meant with "qualified to be integrated into long-term sea ice records" and "The MWRI-ASI SIE can be better integrated into the Sea Ice Index SIE in the Arctic and the OSI-SAF SIE in the Antarctic compared to other products."*
**Reply:**
Thanks for your questions. Because our motivation was changed, these two sentences have been rewritten. Moreover, the similar descriptions were also changed.

Page 1, Line 24-25, we wrote:

Therefore, the MWRI-ASI SIC is comparable with other SIC products and can be applied independently.

Page 22, Line 469, we wrote:

Therefore, the MWRI-ASI SIC is closer to the SSMI-ASI SIC.

Page 22, Line 470-471, we wrote:

The sensitivity of MWRI-ASI SIC to sea ice melting surface is higher than the SSMI-ASI, which suggests the MWRI-ASI SIC product may have better identification ability for the melting surface.

Page 22, Line 472-473, we wrote:

The MWRI-ASI SIE has smaller differences against the Sea Ice Index SIE in the Arctic and the OSI-SAF SIE in the Antarctic compared to other products.

*The ICDC ASI-SSMI version is probably the 5-day median-filtered version? The single day data are available from IFREMER. Both data sets are different in their characteristics. Probably not too much but this should be considered. This is important also for the discussion of the land spillover because the temporal filter has a strong effect. For potential further improvement. I refer to Maaß et al. (2010). Nina Maaß & Lars Kaleschke (2010) Improving passive microwave sea ice concentration algorithms for coastal areas: applications to the Baltic Sea, Tellus A: Dynamic Meteorology and Oceanography, 62:4, 393-410, DOI: 10.1111/j.1600-0870.2009.00452.x*

**Reply:**

Thanks for your question and advice. We have obtained the single-day SSMI-ASI SIC product from IFREMER and compared it to our MWRI-ASI SIC product in Section 4.2. We divided the regions into two parts, one is the region within 50 km away from the coast, and other is the region beyond 50 km away from the coast. Our MWRI-ASI SIC was compared to the single-day SSMI-ASI SIC, five-day SSMI-ASI SIC, AMSR-ASI SIC in these two parts. We also have discussed the land spillover in more details and added this reference.

Page 19-20, Line 403-410, we wrote:

Although some methods have been proposed to solve the land spillover, such as expanding the land mask (Maslanik et al., 1996), subtracting the summer minimum SIC from original images (Cavalieri et al., 1996), and estimating the fraction of land emissivity in the TB (Maaß and Kaleschke, 2010), the SIC differences are still higher in the near-coast regions than in the regions far away from the coast. The SIC MADs between the MWRI-ASI and SSMI-ASI within 50 km away from the coast are larger with values of 6.2% in the Arctic and of 6.7% in the Antarctic, compared to those (3.5% and 5%) beyond 50 km away from the coast. The MADs within 50 km away from the coast between MWRI-ASI SIC and AMSR-ASI SIC are 8.2 % in the Arctic and 8.8% in the Antarctic, which are higher than those (5.5% and 6.5%) beyond 50 km away from the coast.

Page 20, Line 414-415 we wrote:

Due to larger uncertainties of our MWRI-ASI SIC in the near-coast region, it is recommended that the grids extended outward from the coast by 50 km can be removed when using it.

Page 20, Line 416-422 we wrote:

To analyze the influence of temporal filter on land spillover, we acquired the single-day SSMI-ASI SIC product (Single-day SSMI-ASI) from the French Research Institute for Exploitation of the Sea via the Centre d'Exploitation et de Recherche SATellitaire (Ifremer/CERSAT) (Girard-Ardhuin et al., 2008). The SSMI-ASI SIC produced by Hamburg Uni were filtered by a five-day median filter (Five-day SSMI-ASI). In the region within 50 km away from the coast, the SIC MADs between MWRI-ASI and Single-day SSMI-ASI are 5.8% in the Arctic and 6.3% in the Antarctic, which are slightly smaller than those between MWRI-ASI and Five-day SSMI-ASI by 0.4%. It indicates that the SIC uncertainties in the near-coast regions is slightly increased after temporal filter.

---

## Author Comment (AC2)

*In this manuscript, the authors use FengYun satellite series and derived sea ice concentration from 2010-19, which is interesting and valuable to the sea ice community and adds valuable information to the existing SIC climatology from other PMW satellite data. Therefore, I suggest potential publication to ESSD journal. However, I have few major comments that needs to be addressed before publication. Since I am a radar remote sensing scientist with expertise in sea ice (and snow) geophysics, for this round of review, I focus more on the geophysical uncertainty aspect that needs some attention. I am happy to review the revised manuscript and then would give a comprehensive review with more technical and specific comments. Below are my major comments.*

*a) My main concern with your paper is that you have not provided any information on how SNOW as a critical geophyical parameter affects brightness temperature and SIC estimates and its UNCERTAINTY from your datasets. In a warming Arctic and a fluctuating Antarctic, how does shift in sea ice types from MYI to FYI affect snow properties and that in turn affects your SIC estimates? Can you provide a uncertainty range in your derived SICs based on the snow cover and its spatiotemporal variability? For example, FYI cover is characterized by saline snow covers while thicker snow on thinner ice (especially in the Antarctic) is severely affected by flooding, slush and refrozen snow-ice formations. They severely affect the emitting layer correct? I think authors, since they are showing a brand new dataset, its worth and necessary to show the geophysical uncertainties as a quanity. I am a bit disappointed that snow is completely neglected (I should say) in your data product.*

*b) I am really surprised to see almost NO regional variabiity in Antarctic SIC (see Figure 3 (g) to (i)), from your products, when even recent studies (for example: https://essd.copernicus.org/articles/14/619/2022/essd-14-619-2022.html just to quote one) from the same time period as yours have shown large variability in snow depth (that heavily influences SIC) across multiple Antarctic sea ice sectors. This points to my previous comment about accounting for snow depth and its variability in your calculations. I think its a good idea to revisit PMW-derived snow depth data (not that its completely accurate) or any other snow models that can be used to quantify the SIC uncertainty, and combinely use them to figure out the regional SIC variaiblity atleast in the Antarctic. I am sure readers would appreciate that !*

**Reply:**

Thanks for your questions and comments. We have used the PMW-derived snow depth data to estimate overall effects of snow depth on SIC uncertainties in the Arctic and Antarctic. Meanwhile, the effects of snow depth on TB differences were also analyzed. We added the example days in winter and summer for the Antarctic to quantity the SIC uncertainties in more details. In Fig. 3, the spatial distribution of SIC differences do not show the obvious variability with the snow depth, because the SIC differences were calculated from the SIC products all using the ASI algorithm, which have the consistent sensitivity to snow depth.

Page 5, Line 128-132, we wrote:

To quantify the effects of snow depth on SIC uncertainties, we obtained the snow depth on sea ice for the Arctic and Antarctic from the NSIDC (Table 1, AMSR-E version 3, Cavalieri et al., 2014; AMSR2 version 1, Meier et al., 2018). This data is derived from the AMSR TB using the AMSR-E snow-depth-on-sea-ice algorithms and projected on a 12.5-km polar stereographic grid. It is notes that this data is averaged by a five-day running window and only includes the depth of dry snow. Besides, this data provides snow depth for the entire Antarctic, but only for the first-year ice in the Arctic.

Page 12, Line 275-283, we wrote:

When the snow depth is lower than 10 cm, the SIC differences are largest with overall MADs of 6.3% between the MWRI-ASI and SSMI-ASI and of 8.4% between the MWRI-ASI and AMSR-ASI, which are about three times of those (2.2% and 3.2%) when the snow depth is higher than 40 cm. One of the reasons is that the TB differences are also largest when the snow depth is lower than 10 cm, about twice as much as when the snow depth is higher than 10 cm. The spatial distribution of SIC differences do not show the obvious variability with the snow depth (Fig. 3), because all the SIC products are retrieved by the

ASI algorithm, which have the consistent sensitivity to snow depth. In the Antarctic (Fig. 5), when the snow depth is lower than 10 cm, the MAD between MWRI-ASI SIC and SSMI-ASI SIC is 10% on the example day in summer, which are larger than those in winter by 3.5%. Overall, the snow over sea ice has an effect on the SIC uncertainties, which is greater at lower snow depth, especially in summer.

[Figure]

**Figure 5: Frequency of SIC differences between the MWRI-ASI and SSMI-ASI with different snow depths in the Antarctic on 15 February (a) and 15 September (b) 2019. The purple numbers present the MADs in SIC between the MWRI-ASI and SSMI-ASI with the snow depth of 0-10 cm, 10-40 cm, and 40-60 cm.**

Page 22, Line 469-470, we wrote:
Besides, shallower snow depth over sea ice could produce larger SIC uncertainties.

*c) Like reviewer 1 mentioned, the introduction with your rationale is missing or maybe scattered/lost/cluttered within the intro material. I think refining the introduction will help the readability of the paper. Also, just a minor comment. I was curious to know the incidence angle used for the data acquisition. For now I refrain to providing these major comments above. In your revised version, I will provide comprehensive comments on the write up.*

**Reply:**

Thanks for your advice. We have rewritten the introduction. The procedures of our MWRI-ASI SIC product can be seen in the supplement file, including the flow chart, steps, codes, and procedure outcomes. The incidence angle of MWRI sensors is 53, and theirs' characteristics are shown in the following table referring to our previous study: Zhao, X., Chen, Y., Kern, S., Qu, M., Ji, Q., Fan, P., and Liu, Y.: Sea Ice Concentration Derived from FY-3D MWRI and Its Accuracy Assessment, IEEE Trans. Geosci. Remote Sens., 60, https://doi.org/10.1109/TGRS.2021.3063272, 2022,

TABLE I
Main Characteristics of MWRI, AMSR2, and SSMIS

| Sensor | MWRI | | AMSR2 | | SSMIS | |
|---|---|---|---|---|---|---|
| Satellite | FY-3D | | GCOM-W1 | | DMSP F-18 | |
| Altitude (km) | 836 | | 700 | | 850 | |
| Incidence Angle (°) | 53 | | 55 | | 53 | |
| Equator Crossing Time (Local Time Zone) | A: 14:00 D: 02:00 | | A: 13:30 D: 01:30 | | A: 18:03 D: 07:08 | |
| Swath Width (km) | 1400 | | 1450 | | 1707 | |
| Scan Period (s) | 1.8 | | 1.5 | | 1.9 | |
| Number of Scans | 266 | | 243 / 486 | | 256 | |
| Antenna Size (m) | 0.977×0.897 | | 2 | | 0.61 | |
| Center Frequency (GHz) Polarization | 10.65 | V/H | 10.65 | V/H | – | – |
| | 18.7 | V/H | 18.7 | V/H | 19.35 | V/H |
| | 23.8 | V/H | 23.8 | V/H | 22.235 | V |
| | 36.5 | V/H | 36.5 | V/H | 37 | V/H |
| | 89 | V/H | 89 | V/H | 91.655 | V/H |
| Footprint Size Along Scan × Across Scan (km²) Sampling Interval Along Scan × Across Scan (km²) | 51×85 | / 6×12 | 24×42 | / 10×10 | – | – |
| | 30×50 | / 6×12 | 14×22 | / 10×10 | 42×70 | / 25×25 |
| | 27×45 | / 6×12 | 11×10 | / 10×10 | 42×70 | / 25×25 |
| | 18×30 | / 6×12 | 7×12 | / 10×10 | 28×45 | / 25×25 |
| | 9×15 | / 6×12 | 3×5 | / 5×5 | 13×15 | / 12.5×12.5 |

"A" is ascending orbit and "D" is descending.

---

## Referee Report (RR1)

Review 17 February 2023 Earth System Science Data

A new sea ice concentration product in the polar regions derived from the FengYun-3 MWRI sensors Chen et al.

This paper describes a new sea ice concentration (SIC) data product produced using a modified version of the ASI algorithm and a new intercalibrated brightness temperature data set from the FengYun-3 series of satellite Microwave Radiation Imager. The paper provides a thorough intercomparison of the new SIC product with similar passive microwave SIC products from AMSR-E/AMSR2 and SSMI/SSMIS as well as a comparison of the derived sea ice extents to other existing sea ice extent products. In general, the new dataset compares similarly to the other products and the authors do a nice job of framing their new data set as an independent measure of SIC while new sensors for the other SIC products come online in the future. The paper also describes the common limitations of their data that affect all passive microwave sea ice concentration products. The dataset is publicly available at the PANGAEA repository, and I was able to download and read a sample of the data without any problems.

I think this work provides a good contribution and is appropriate for publication in ESSD pending a few minor revisions as described in my comments below.

**Minor Comment**

I have one minor comment regarding the intercomparison of the gridded SIC product with shipbased observations discussed in Section 3.3. Specifically, I disagree that larger disagreement between the satellite SIC and ship observations in the low SIC categories automatically means that the satellite SIC is less accurate. There is a more complex relationship here related to the distribution of sea ice within the gridded cell versus an observation at a single point or along a transect. I suggest the authors add a bit more discussion about how the differences in the scale of these two observation types are not a one-to-one comparison. The remainder of my comments listed below are very minor technical suggestions.

Technical Comments

L64: Change "easily ignored by the PM observations" to "unresolved by the PM observations"

L73: Grammar - change "which is only use" to "which only uses"

L131: Change "It is notes" to "It is noted"

L132: I recommend deleting "Besides" from this sentence.

Figure 7: The color scale below the figure does not seem to map to the data in any regularly spaced bins, other than dark blue = 0, white or grey = 3, and dark red = 55. This makes some lower percentage proportions (e.g., 5 - 15 range) appear to be more significant (pink and red)

than they may be. I suggest binning the data into a regularly spaced color scale and not using a diverging color map.

L345: An observer on a ship is reporting the concentration of sea ice immediately surrounding the ship, not the concentration of ice distributed around the full area of a grid cell. Wouldn't the bigger differences between MWRI-ASI SIC and the ship observations in the lower sea ice concentration groups be related to differences in the distribution of sea ice within the gridded observations versus a point observation from a ship? I think the conclusion at line 345 that MWRI-ASI SIC is more accurate in high concentration regions is more complex than stated.

L375 and 376: Typo? Should "lightly" be "slightly"?

L416 – 422: Do not introduce a new dataset and results in the discussion section. I suggest moving this up to the results section.

L428-430: Is this future work? "in the next step" suggests that you will address it in the next section of the paper. I recommend changing the wording here to say this will be future work.

L455: I recommend changing "Besides" to "Additionally" to make it clearer that these biases are also provided with the SIC data set.

---

## Author Response (AR2)

Dear Dr. Petra Heil,

Many thanks for the review of our manuscript. We have considered each comment from the reviewers carefully and provided an itemized response to the comments as follows. We also polished the language. Attached is a revised version of our manuscript, in which we marked all the changes in blue. Also, the revised version of supplement file is attached.

Looking forward to hearing from you soon.

Best Regards,
Xiaoping Pang and other contributors.

**Response to comments from Vishnu Nandan**

*Dear Chen et al. Thanks for answering my questions on your manuscript and incorporating my suggestions in your revised manuscript. Its much improved. However, I found your description of Figure 5 (lines 275 to 283 in your revised manuscript) to be vague and sort of 'washed off' the main points.*

*a)    Why is the TB difference greater for thin snow covers compared to thick?*

**Reply:**

Thanks for your question. We have added the statistic results of TB differences with different snow depths, shown in the supplement file (Table S3). Results show that the TB differences are larger for thin snow covers compared to thick snow covers. Additionally, the statistic results of SIC differences with different snow depths were shown in Table S2 of the supplement file.

Page 12-13, Line 278-282, we wrote:

When the snow depth is lower than 10 cm, the SIC differences are the largest with a mean MAD of 7% between the MWRI-ASI and SSMI-ASI and of 9% between the MWRI-ASI and AMSR-ASI, which are about three times of those (2% and 3%) when the snow depth is higher than 40 cm (Table S2 of the supplement file). One of the reasons is that the TB differences are also largest when the snow depth is lower than 10 cm, about twice as much as when the snow depth is higher than 10 cm (Table S3 of the supplement file).

Supplement file, Page 8, we wrote:

**Table S2: MADs between the MWRI-ASI SIC and other two SICs with the snow depth of 0-10 cm, 10-40 cm, and 40-60 cm in the Arctic and Antarctic from 2010 to 2019.**

|  | Arctic (%) | | | Antarctic (%) | | |
|---|---|---|---|---|---|---|
|  | 0-10 cm | 10-40 cm | 40-60 cm | 0-10 cm | 10-40 cm | 40-60 cm |
| SSMI-ASI | 7 | 2 | 2 | 6 | 3 | 2 |
| AMSR-ASI | 9 | 4 | 3 | 8 | 5 | 3 |

**Table S3: MADs in the polarization difference at 89 GHz between the MWRI sensor and other two PM series sensors with the snow depth of 0-10 cm, 10-40 cm, and 40-60 cm in the Arctic and Antarctic from 2010 to 2019**.

|  | Arctic (K) | | | Antarctic (K) | | |
|---|---|---|---|---|---|---|
|  | 0-10 cm | 10-40 cm | 40-60 cm | 0-10 cm | 10-40 cm | 40-60 cm |
| SSMI | 2.7 | 1.3 | 1.2 | 2.6 | 1.5 | 1.3 |
| AMSR | 2.4 | 1.1 | 1.0 | 2.3 | 1.3 | 1.0 |

*b) Also, you make a speculative statement about Tb differences being high for thin snow covers in summer. What is the reason for that? Is it geophysical? This chimes back to my comments from my first round where especially in the Antarctic, there is stronger effect of snow (due to loading, flooding, melt/refreeze cycles and its deterrent effects such as slush/snow-ice/ice layers etc etc). But I dont think the authors have addressed any of them or even taken into consideration. Yes, this is a paper focusing on the product. But at the same time, there should be at least a discussion on geophysical uncertainties especially with snow being the dominant parameter affecting your SIC. I still feel that needs to be accounted for in your revised manuscript.*

**Reply:**

Thanks for your questions and comments. We have reconsidered the influence of snow on SIC. We suggested this phenomenon can be explained by metamorphoses in the properties of snow during summer, which is supported by some references. Also, we have presented the explanation in the text.

Page 13, Line 285-296, we wrote:

This could be explained by metamorphoses in the properties of snow over sea ice during summer, such as increased wetness (even saturated with meltwater), increased snow density, increased snow grain size, the occurrence of diurnal melt–refreeze cycles on surface, and slush on surface, etc., which have an impact on TB (Ivanova et al., 2015; Kern et al., 2016, 2019). The increase of snow wetness usually leads to an increase in TB of about 10 - 60 K, while the increase of snow grain size, which would cause the geophysical properties of snow cover to be very close to the surface scattering layer of sea ice, typically leads to a decrease in TB of about 15 - 35 K, resulting in large uncertainty of SIC (Kern et al., 2016). With the increase in snow depth, e.g., > 40 cm, the corresponding increased snow load may lead to a negative ice freeboard, especially for the thin ice in the Antarctic, resulting in the slush layer appearing between the snow cover and the ice layer (Li et al., 2023). However, such slush layer is often thin, and the surface covered with thick snow would generally keep dry. This mechanism can be used to explain why the deviation of SIC is always the smallest for thick snow cover in both winter or summer. Thus, the snow over sea ice could play a significant role on the SIC uncertainties, which is greater at lower snow depth, especially in summer.

*I think the paper would be more impactful if you could answer my a) and b). The paper is good to be published subject to this revision (which wont take much of your time, if you can find some literature discussing my above mentioned issues). Think about it :)*

**Response to comments from referee**

*This paper describes a new sea ice concentration (SIC) data product produced using a modified version of the ASI algorithm and a new intercalibrated brightness temperature data set from the FengYun-3 series of satellite Microwave Radiation Imager. The paper provides a thorough intercomparison of the new SIC product with similar passive microwave SIC products from AMSRE/AMSR2 and SSMI/SSMIS as well as a comparison of the derived sea ice extents to other existing sea ice extent products. In general, the new dataset compares similarly to the other products and the authors do a nice job of framing their new data set as an independent measure of SIC while new sensors for the other SIC products come online in the future. The paper also describes the common limitations of their data that affect all passive microwave sea ice concentration products. The dataset is publicly available at the PANGAEA repository, and I was able to download and read a sample of the data without any problems.*

*I think this work provides a good contribution and is appropriate for publication in ESSD pending a few minor revisions as described in my comments below.*

*Minor Comment*
*I have one minor comment regarding the intercomparison of the gridded SIC product with ship-based observations discussed in Section 3.3. Specifically, I disagree that larger disagreement between the satellite SIC and ship observations in the low SIC categories automatically means that the satellite SIC is less accurate. There is a more complex relationship here related to the distribution of sea ice within the gridded cell versus an observation at a single point or along a transect. I suggest the authors add a bit more discussion about how the differences in the scale of these two observation types are not a one-to-one comparison. The remainder of my comments listed below are very minor technical suggestions.*

**Reply:**
Thanks for your comment and suggestion. We have added the discussion about the differences in the scales between the gridded SIC products and ship-based observations in discussion Section 4.3.

Page 23, Line 466-474, we wrote:
This study and the results given by Spreen et al., (2008) both presented that the SIC differences between PM-based and ship-based observations are larger in the low-SIC region than those in the high-SIC region. The large SIC differences can be explained by the different spatial and temporal scales between PM SICs and ship-based SICs (Beitsch et al., 2015; Kern et al., 2019). Ship-based SICs are obtained on an elliptically shaped area of 1 km on each side of the ship, while the footprint sizes of PM frequencies were considerably larger than 1 km, which are several kilometers to tens of kilometers. In contrast to ship-based SIC gained by observers at a specific time, the PM SICs are the daily averages combined with swath SICs from different time in one calendar day. The ship-based SIC may not be fully representative of the entire grid of PM SIC and the observation results may also be affected by visibility and light around the ship.

*Technical Comments*
*L64: Change "easily ignored by the PM observations" to "unresolved by the PM observations"*
**Reply:**
Thanks for your suggestion. We have changed it.

Page 2, Line 63-65, we wrote:
Results indicated that the three SIC products were slightly lower than the ship-based SIC in winter but higher in summer by 10% to 12%, because the small-scale morphological features such as leads and sparse small floes are unresolved by the PM

observations.

*L73: Grammar - change "which is only use" to "which only uses"*

**Reply:**

Thanks for your suggestion. We have corrected it.

Page 3, Line 73-75, we wrote:

Due to low frequencies applied in the NT2 algorithm, the original resolutions of the NT2 SIC products are lower than those of the ASI SIC products, which only uses the highest frequency with high spatial resolution (Spreen et al., 2008).

**Reply:**

Thanks for your suggestion. We have changed it.

Page 5, Line 132, we wrote:

It is noted that this data is averaged by a five-day running window and only includes the depth of dry snow.

*L132: I recommend deleting "Besides" from this sentence.*

**Reply:**

Thanks for your suggestion. We have deleted it.

Page 5, Line 133, we wrote:

This data provides snow depth for the entire South Ocean in the Antarctic, but only for the first-year ice in the Arctic.

*Figure 7: The color scale below the figure does not seem to map to the data in any regularly spaced bins, other than dark blue = 0, white or grey = 3, and dark red = 55. This makes some lower percentage proportions (e.g., 5 – 15 range) appear to be more significant (pink and red) than they may be. I suggest binning the data into a regularly spaced color scale and not using a diverging color map.*

**Reply:**

Thanks for your suggestion. We have changed color scale of Figure 7.

Page 17, Line 368, we wrote:

[Figure]

**Figure 7: Proportion of data pairs (number in the grid) of the PM SIC products vs SIC differences between the PM SIC products and ship-based SIC. The PM SIC products are divided to 15–30%, 30–70%, and 70–100% (horizontal axis). The SIC differences are grouped with an interval of 20% from -100% to 100% (vertical axis).**

*L345: An observer on a ship is reporting the concentration of sea ice immediately surrounding the ship, not the concentration of ice distributed around the full area of a grid cell. Wouldn't the bigger differences between MWRI-ASI SIC and the ship observations in the lower sea ice concentration groups be related to differences in the distribution of sea ice within the gridded observations versus a point observation from a ship? I think the conclusion at line 345 that MWRIASI SIC is more accurate in high concentration regions is more complex than stated.*

**Reply:**

Thanks for your question and advice. We have deleted the sentences at original line 345 and added the discussion in Section 4.3.

Page 23, Line 466-474, we wrote:

This study and the results given by Spreen et al., (2008) both presented that the SIC differences between PM-based and ship-based observations are larger in the low-SIC region than those in the high-SIC region. The large SIC differences can be explained by the different spatial and temporal scales between PM SICs and ship-based SICs (Beitsch et al., 2015; Kern et al., 2019). Ship-based SICs are obtained on an elliptically shaped area of 1 km on each side of the ship, while the footprint sizes of PM frequencies were considerably larger than 1 km, which are several kilometers to tens of kilometers. In contrast to ship-based SIC gained by observers at a specific time, the PM SICs are the daily averages combined with swath SICs from different time in one calendar day. The ship-based SIC may not be fully representative of the entire grid of PM SIC and the observation results may also be affected by visibility and light around the ship.

*L375 and 376: Typo? Should "lightly" be "slightly"?*

**Reply:**

Thanks for your advice. We have corrected it.

Page 19, Line 394-396, we wrote:

Overall, the polarization difference at 89 GHz of the MWRI sensor is slightly higher than that of the SSMI sensor with a mean positive bias of 0.9 K and slightly lower than that of the AMSR sensor with a mean negative bias of -1.1 K.

*L416 – 422: Do not introduce a new dataset and results in the discussion section. I suggest moving this up to the results section.*

**Reply:**

Thanks for your suggestion. At original line 416 to 422 (now line 433 to 439), we aimed to discuss the influence of temporal filter on land spillover. In the result Section 3.1, we found that our MWRI-ASI SIC is limited in terms of the land spillover. For a better discussion of land spillover, the second paragraph of the discussion Section 4.2 illustrated this limitation in more details. We added the Single-day SSMI-ASI SIC product only to analyze the land spillover. The Single-day SSMI-ASI SIC product are not added into the analysis of the result Section. Therefore, we think these sentences are better placed in the discussion Section.

*L428-430: Is this future work? "in the next step" suggests that you will address it in the next section of the paper. I recommend changing the wording here to say this will be future work.*

**Reply:**

Thanks for your suggestion. We have corrected it.

Page 21, Line 447-449, we wrote:

Thus, in the future work, we will attempt to identify and remove the spurious ice caused by land spillover and weather effects, by combining the optical or synthetic aperture radar images with higher resolutions, to further improve our MWRI-ASI SIC product.

*L455: I recommend changing "Besides" to "Additionally" to make it clearer that these biases are also provided with the SIC data set.*

**Reply:**

Thanks for your suggestion. We have changed it.

Page 23, Line 482-483, we wrote:

Additionally, the biases between this SIC dataset and other two ASI SIC products, i.e., SSMI-ASI and AMSR-ASI, are provided.

---

## Author Response (AR3)

Dear Ken Mankoff,

Many thanks for the review of our manuscript. We have considered each comment carefully and provided an itemized response to the comments as follows. Since there was no option to upload track changes in the system this time, we provided the track changes in the itemized response, in which the changes were marked in blue.

Looking forward to hearing from you soon.

Best Regards,
Ying Chen and other contributors.

**Response to comments**

*This paper has a lot of acronyms. I think it should have a List of Acronyms or Glossary somewhere to help the reader. I note that see "GR" is never defined.*

**Reply:**

We have added a table of abbreviations in the section "Appendix A: Abbreviations". We moved the definition of GR to where it first appeared and gave its full term.

Page 6, Line 154-155, we wrote:

The gradient ratio (GR) is defined as the TB difference with V-polarization between high and low frequencies over the sum of these two TB.

Page 24-25, we wrote:

**Table A1. List of abbreviations used in the paper.**

| Abbreviation | Term |
| --- | --- |
| SIC | Sea ice concentration |
| SIE | Sea ice extent |
| PM | Passive microwave |
| MWRI | Microwave Radiation Imager |
| FY-3 | FengYun-3 |
| SSMI | Special Sensor Microwave Imager series |
| SMMR | Scanning Multichannel Microwave Radiometer |
| SSM/I | Special Sensor Microwave/Imager |
| SSMIS | Special Sensor Microwave Imager Sounder |
| AMSR | Advanced Microwave Scanning Radiometer series |
| AMSR2 | Advanced Microwave Scanning Radiometer 2 |
| AMSR-E | Advanced Microwave Scanning Radiometer-EOS |
| MIZ | Marginal ice zone |
| PIZ | Pack ice zone |
| ASI | Arctic Radiation and Turbulence Interaction Study Sea Ice |
| BST | Bootstrap |

| | |
|---|---|
| NT2 | Enhanced NASA Team |
| NT | NASA Team |
| PMA | Passive microwave algorithm |
| NSMC | Chinese National Satellite Meteorological Center |
| OSI-SAF | Ocean and Sea Ice Satellite Application Facility |
| NSIDC | National Snow and Ice Data Center |
| ICDC | Integrated Climate Data Center |
| GESR | Goddard Earth Science Research |
| TB | Brightness temperature |
| H | Horizontal polarization |
| V | Vertical polarization |
| GR | Gradient ratio |
| Ice Watch/ASSIST | Ice Watch/Arctic Ship-based Sea-Ice Standardization |
| ASPeCt | Antarctic Sea Ice Processes and Climate |
| CHINARE | Chinese National Arctic and Antarctic Research Expedition |
| MAD | Mean absolute deviation |
| RMSD | Root mean standard deviation |
| $R$ | Correlation coefficient |

*Data is provided as Float32 but is only values -2, -1, and 0-100. I assume integer precision is good enough. Therefore, if it were Integer data it would be 1/4th the size. I suggest re-uploading compressed GeoTIFFs of 8-bit signed integer values.*

**Reply:**

We have updated the GeoTIFFs to 8-bit signed integer values. However, this dataset has been accepted for one year. We have tried contacting the editor of data but have not received a response. Therefore, it is difficult to re-uploading the updated data.

*31: Correct "The PM SIC is the most vital data" to "The PM SIC is the most vital dataset".*

**Reply:**

We have corrected it.

Page 1, Line 31-32, we wrote:

The PM SIC is the most vital dataset to initialize the sea ice condition for climate modeling due to its continuous observations (Meier, 2019).

*42: Change "are in planning stage." to "are in planning or concept stage (esastar, accessed 2023-06-10)."*

**Reply:**

We have changed it.

Page 2, Line 40-42, we wrote:

The new missions for successors of these two sensors or the launch plans for other instruments, e.g., the Copernicus Imaging Microwave Radiometer (Jiménez et al., 2021) and Weather Satellite Follow–On–Microwave (Newell et al., 2020), are in planning or concept stage (Esastar, accessed 2023-06-10).

*46ff: Shorten to "SSMIS data have been used to bridge the 2011 to 2012 data gap between Advanced Microwave Scanning Radiometer-EOS (AMSR-E) and AMSR2, to generate a continuou time series (Meier and Ivanoff, 2017)."*

**Reply:**

We have shortened it. However, Meier and Ivanoff, (2017) did not generate a continuous time series. They just used SSMIS data as a bridge to compare AMSR-E total sea ice extent in 2010 with AMSR2 total sea ice extent in 2013. Therefore, we further corrected this sentence.

Page 2, Line 46-48, we wrote:

SSMIS data have been used to bridge the 2011 to 2012 data gap between Advanced Microwave Scanning Radiometer-EOS (AMSR-E) and AMSR2 (Meier and Ivanoff, 2017).

*48ff: Remove "An updated SIC product from the MWRI sensors after systematic assessment may be used to verify the SIC products derived from the next generations of PM sensor."*

**Reply:**

We have removed it.

*55ff: Change "in the melting seasons than in the freezing seasons." to "during sea-ice melt rather than during freezing conditions."*

**Reply:**

We have changed it.

Page 2, Line 54-55, we wrote:

Thus, the uncertainties of SIC and SIE are generally greater during sea-ice melt rather than during freezing conditions.

*79: Correct "were used" to "are used".*

**Reply:**

We have corrected it.

Page 3, Line 77-78, we wrote:

The recently re-calibrated brightness temperature (TB) of the MWRI sensors provided by NSMC are used in this study to ensure the consistency of this new MWRI SIC product.

*88: Correct "More details" to "Further details" and "were given" to "are given".*

**Reply:**

We have corrected it.

Page 3, Line 87, we wrote:

Further details of the MWRI characteristics are given in Zhao et al. (2022).

*86: Correct "in conically scanning mode" to "in conical scanning mode".*

**Reply:**

We have corrected it.

Page 3, Line 85-86, we wrote:

The MWRI sensors measure the radiation of the land, ocean, and atmosphere in conical scanning mode at five frequencies between 10 to 89 GHz at both horizontal (H) and vertical (V) polarization.

*90: Remove "still".*

**Reply:**

We have removed it.

Page 3, Line 88-89, we wrote:

Although the MWRI sensors onboard the different FY-3 satellites have consistent technical characteristics, the TB data obtained from different MWRI sensors reveal some deviations.

*90: Change "were re-calibrated" to "are re-calibrated".*
*91: Change "focused" to "focus".*

**Reply:**

We have changed them.

Page 3, Line 89-91, we wrote:

Therefore, the MWRI TB data are re-calibrated using the operational algorithm, which focus on the hot load, antenna, and receiver calibration, reducing the TB deviations of different MWRI sensors.

*93: Change "This study used" to "This study uses".*
*93: Remove "which are" to shorten.*

**Reply:**

We have changed and removed them.

Page 3, Line 92-93, we wrote:

This study uses the re-calibrated level 1 swath MWRI TB data from the FY-3B, FY-3C, and FY-3D satellites, provided by the NSMC and available at http://www.richceos.cn (Table 1).

*94: Change "better performance" to "improved performance".*
*95: Change "than others" to "compared to its predecessors".*

**Reply:**

We have changed them.

Page 3, Line 93-94, we wrote:

Considering the improved performance of the FY-3D MWRI sensor compared to its predecessors, we preferentially selected the MWRI TB from the FY-3D, followed by the FY-3C and FY-3B.

*105: Remove "km" from "12.5-km".*

**Reply:**

We have removed it.

Page 4, Line 103-104, we wrote:

This SSMI TB product is projected on 12.5- and 25-km polar stereographic grids at high and low frequencies, respectively.

*106: Change "The time coverage" to "The temporal coverage".*

**Reply:**

We have changed it.

Page 4, Line 105-106, we wrote:

The temporal coverage of these two daily TB products is corresponding to that of the MWRI TB.

*114: Remove "sensors" from "series sensors".*
*115: Remove "sensors" from "series sensors".*
*116ff: Remove "produced by the Institute of Environmental Physics (IUP) of the University of Bremen".*

**Reply:**

We have removed them.

Page 4, Line 112-115, we wrote:

One is available from the Integrated Climate Data Center (ICDC) of the University of Hamburg, which is derived from the Special Sensor Microwave Imager series projected onto a 12.5-km polar stereographic grid (SSMI-ASI) (Kern et al., 2023). The other is derived from the Advanced Microwave Scanning Radiometer series projected onto a 6.25-km polar stereographic grid (version 5.4, AMSR-ASI) (Melsheimer and Spreen, 2023).

**Reply:**

We have changed it.

Page 5, Line 119-121, we wrote:

To evaluate differences in the uncertainties of SIC between the melting and freezing periods, we use the data of Arctic sea ice surface melt or freeze onset to define the ice melting and freezing periods, which (version 371s, Table 1) are available from the Goddard Earth Science Research (https://earth.gsfc.nasa.gov/index.php/cryo/data).

**Reply:**

We have corrected it.

Page 5, Line 121-123, we wrote:

These data are obtained from the SSMI series sensors using the passive microwave algorithm (PMA) projected onto a 25-km polar stereographic grid, which includes the onsets of the early melt, melt, freeze, and late freeze for the sea ice surface (Markus et al., 2009).

**Reply:**

We have changed them.

Page 5, Line 130-132, we wrote:

To assess the performance of different SIC products in the MIZ, where the accuracy of PM SIC is generally low, the monthly MIZ SIE and MIZ SIE fraction (the ratio between the MIZ SIE and the total SIE) obtained from the three ASI SIC products

are compared.

*134: Change "resampled" to "resample".*
*135: Change "during the entire overlap periods," to "for the entire instrument overlap".*
**Reply:**
We have changed them.

Page 5, Line 132-134, we wrote:
We resample the AMSR-ASI SIC onto a 12.5-km grid to match the MWRI-ASI SIC and SSMI-ASI SIC to compare the SIC for the entire instrument overlap, as well as the winter (Arctic: December – May, Antarctic: June – November) and summer months (Arctic: June – November, Antarctic: December – May), respectively.

*144/145: Change "SIC product developed by the Ocean and Sea Ice Satellite Application Facility Norwegian Meteorological Institute (version 2, OSI-SAF) (Lavergne et al., 2020)." to "SIC product described in Lavergne et al. (2020)."*
**Reply:**
We have changed it.

Page 5, Line 141-142, we wrote:
The fourth SIE product (called as OSI-SAF) is derived from the SIC product described in Lavergne et al. (2020).

*146: Change "were quantified" to "are quantified".*
*146: Change "their entire overlap periods"their to "their entire overlap periods".*
**Reply:**
We have changed them. The second change is confusing. As the previous comments "135: Change 'during the entire overlap periods,' to 'for the entire instrument overlap'.", we followed this comment to change this sentence.

Page 5, Line 143-144, we wrote:
The differences between the MWRI-ASI SIE and four existing SIE products are quantified for the entire instrument overlap, as well as the winter and summer months.

*147: Change "were compared" to "are compared".*
**Reply:**
We have changed it

Page 5, Line 144-145, we wrote:
The 2010–2019 trends of the MWRI-ASI SIE and four existing SIE products are compared.

*153: Change "In the previous study" to "In a previous study".*
**Reply:**
We have changed it.

Page 6, Line 150-151, we wrote:
In a previous study (Zhao et al., 2022), a TB bias-correction was performed to reduce the bias between daily MWRI TB and AMSR2 TB.

*156: Change "weather filters" to "weather filters [GR(x/y)]".*

**Reply:**

We have changed it.

Page 6, Line 153-154, we wrote:

Two weather filters, i.e., GR(36.5/18.7) and GR(23.8/18.7), and a monthly maximum ice extent mask were utilized to remove the spurious sea ice.

*157: Change "More details" to "Details".*
*157: Change "were given in" to "are given in".*

**Reply:**

We have changed them.

Page 6, Line 155-157, we wrote:

Details about the dynamic tie points ASI algorithm are given in Zhao et al. (2022), and the details about the ASI algorithm can be referred to Svendsen et al. (1987), Kaleschke et al. (2001) and Spreen et al. (2008).

*164/caption Tab.2: Change "Differences in the parameters" to "Parameters".*

**Reply:**

We have changed it.

Page 6, Line 163, we wrote:

Table 2. Parameters or operations used in the previous and modified algorithms.

*166: Change "would cause" to "gives rise".*

**Reply:**

We have changed it.

Page 6, Line 165, we wrote:

Zhao et al. (2022) directly used the tie points from the AMSR series to initiate the SIC derivation, which gives rise large bias in initial SIC due to differences between MWRI and AMSR TB.

*171: Change "referential SIC" to "reference SIC".*

**Reply:**

We have changed it.

Page 6, Line 169-170, we wrote:

We used daily SSMI-ASI SIC in 2018 as the reference SIC and computed the daily average of the MWRI-ASI SIC and SSMI-ASI SIC (SIC > 15%).

*171/172: Change "Then, the linear regression was conducted between" to "The next step involves the linear regression between".*

**Reply:**

We have changed it.

Page 6, Line 170-171, we wrote:

The next step involves the linear regression between the daily average MWRI-ASI SIC and referential SSMI-ASI SIC.

*172/173: Change "As initial tie points we selected the pair satisfying requirements with the slope closer to 1" to "As initial tie point the pair satisfying requirements with the slope closer to 1 was selected".*

**Reply:**

We have changed this sentence.

Page 6, Line 171-173, we wrote:

As initial tie point the pair satisfying requirements with the slope closer to 1, intercept closer to 0, and relatively low standard deviation (Std) was selected from the 6000 samples.

*179: Change "Antarctic tie points were generated by the following procedures." to "process for Antarctic tie points is outlined next."*

**Reply:**

We have changed it.

Page 7, Line 177-178, we wrote:

We adopted the dynamic tie points proposed by Zhao et al. (2022) in Arctic, and process for Antarctic tie points is outlined next.

*179: Shorten "The conditions of sea ice tie-point samples in Antarctic are defined as follows:" to "Antarctic sea ice tie points are derived as".*

*180/181: Shorten "The conditions of open water tie-point samples in Antarctic are defined as follows:" to "Antarctic open water tie points are derived as".*

**Reply:**

We have shortened them.

Page 7, Line 178-181, we wrote:

Antarctic sea ice tie points are derived as: the initial SICs of grids are larger than 95%, and the grids are within the monthly minimum ice extent and 100 km away from the coast. Antarctic open water tie points are derived as: the initial SICs of grids fall within the range [-10%, 10%], and the grids are away from the monthly ice edge by 200–350 km and away from the coast by 100 km.

*182/183: Change "The Section S1.4 of the supplement file illustrates detailed procedures of generating the dynamic tie points." to "Details on generating the dynamic tie points are given in Section S1.4".*

**Reply:**

We have changed it.

Page 7, Line 181, we wrote:

Details on generating the dynamic tie points are given in Section S1.4.

*184: Change "were determined" to "are determined".*

**Reply:**

We have changed it.

Page 7, Line 182-183, we wrote:

The thresholds of the weather filters GR(36.5/18.7) and GR(23.8/18.7) are determined as 0.045 and 0.04 by Zhao et al. (2022), respectively, as the AMSR series sensors.

*186: Remove "can".*

**Reply:**

We have removed it.

Page 7, Line 183-185, we wrote:

In this study, we chose 0.05 and 0.045 as thresholds of GR(36.5/18.7) and GR(23.8/18.7), respectively, as the SSMI series sensors, which generally remove the weather effects (Gloersen and Cavalieri, 1986; Cavalieri et al., 1995).

*188: Replace "against to" with "over".*

**Reply:**

We have replaced it. Due to the comments of "'GR' is never defined" and of "Change 'weather filters' to 'weather filters [GR(x/y)]'.", we moved this sentence about the definition of GR to where it first appeared.

Page 6, Line 154-155, we wrote:

The gradient ratio (GR) is defined as the TB difference with V-polarization between high and low frequencies over the sum of these two TB.

*196/197: Would be good to list the specific databases by DOI from IceWatch and Pangaea.*

**Reply:**

We have listed the specific databases by DOI from Pangaea. However, the specific databases from IceWatch do not give the DOI, so we did not add them.

Page 7, Line 190-195, we wrote:

To validate the accuracy of the MWRI-ASI SIC, we collected the observational SIC from various ship-based measurement programs of the Chinese National Arctic and Antarctic Research Expedition (CHINARE) conducted by the Polar Research Institute of China (Lei et al., 2017) and a standardized ship-based observation dataset (ESA-SICCI) produced by Kern (2019), as well as those available in the IceWatch (https://icewatch.met.no/cruises) and PANGAEA databases (https://www.pangaea.de) (Arndt, 2019; Arndt and van Caspel, 2017; Katlein et al., 2014; Arndt, 2018; Arndt and Castellani, 2019; Hendricks et al., 2012).

Page 26, Line 528-537, we wrote:

Arndt, S.: Sea ice conditions during POLARSTERN cruise PS111 (ANT-XXXIII/2, FROST), Alfred Wegener Institute, Helmholtz Centre for Polar and Marine Research, Bremerhaven, PANGAEA, https://doi.org/10.1594/PANGAEA.887697, 2018.

Arndt, S.: Sea ice conditions during POLARSTERN cruise PS118 (LARSEN), Alfred Wegener Institute, Helmholtz Centre for Polar and Marine Research, Bremerhaven, PANGAEA, https://doi.org/10.1594/PANGAEA.901263, 2019.

Arndt, S. and van Caspel, M.: Sea ice conditions during POLARSTERN cruise PS103 (ANT-XXXII/2), Alfred Wegener Institute, Helmholtz Centre for Polar and Marine Research, Bremerhaven, PANGAEA, https://doi.org/10.1594/PANGAEA.880046, 2017.

Arndt, S. and Castellani, G.: Sea ice conditions during POLARSTERN cruise PS117, Alfred Wegener Institute, Helmholtz Centre for Polar and Marine Research, Bremerhaven, PANGAEA, https://doi.org/10.1594/PANGAEA.901279, 2019.

Page 27, Line 573-575, we wrote:

Hendricks, S., Nicolaus, M., and Schwegmann, S.: Sea ice conditions during POLARSTERN cruise ARK-XXVII/3 (IceArc), Alfred Wegener Institute, Helmholtz Centre for Polar and Marine Research, Bremerhaven, PANGAEA, https://doi.org/10.1594/PANGAEA.803221, 2012.

Page 28, Line 587-589, we wrote:

Katlein, C., Arndt, S., and Nicolaus, M.: Sea ice conditions during POLARSTERN cruise PS86 (ARK-XXVIII/3 AURORA), Alfred Wegener Institute, Helmholtz Centre for Polar and Marine Research, Bremerhaven, PANGAEA, https://doi.org/10.1594/PANGAEA.835578, 2014.

*213/Caption Fig.1: Correct "statistical significant" to "statistical significance".*

**Reply:**

We have corrected it.

Page 8, Line 210-212, we wrote:

The statistical significance at 95% and 99% confidence levels are marked by * and **, respectively, and those below 95% confidence level are not marked, the same below.

*216: Correct "Comparisons" to "Comparison".*
*298: Correct "Comparisons" to "Comparison".*

**Reply:**

We have corrected them.

Page 8, Line 214, we wrote:

3.1 Comparison of the ASI SIC products

Page 13, Line 291, we wrote:

3.2 Comparison of the SIE products

*304: Replace "decreasing trend" with "reduction".*
*305: Replace "decreasing trends" with "reductions".*

**Reply:**

We have replaced them.

Page 14, Line 301-303, we wrote:

In the Antarctic, the largest reduction is identified for the MWRI-ASI SIE ($-191,993$ km$^2$ yr$^{-1}$, $P<0.01$), and the difference in trends between the MWRI-ASI SIE and OSI-SAF SIE is the lowest ($-8,928$ km$^2$ yr$^{-1}$ or about 5%).

Page 14, Line 304-305, we wrote:

In the period from 1979 to 2019, the four existing SIE products show significant reductions (about -55,000 km$^2$ yr$^{-1}$, $P<0.01$) in the Arctic and increasing trends (about 7,500 km$^2$ yr$^{-1}$, $P<0.01$) in the Antarctic (Table 4).

*341/Caption Tab.4: Add "The statistically significant at 95% and 99% confidence levels are marked by \* and \*\*, respectively."*

**Reply:**

We have added it. Due to the comment "213/Caption Fig.1: Correct 'statistical significant' to 'statistical significance'.", we further corrected this sentence.

Page 15, Line 337, we wrote:

The statistical significance at 95% and 99% confidence levels are marked by * and **, respectively.

*348: Correct "Comparisons to the" to "Comparison with".*

**Reply:**

We have corrected it.

Page 16, Line 346, we wrote:

3.3 Comparison with ship-based SIC

*356/357: Shorten "This implies that the MWRI-ASI SIC has a better accuracy in winter than in summer, which is due" to "Higher accuracy of the MWRI-ASI SIC during winter compared to during summer is due".*

**Reply:**

We have shortened it.

Page 16, Line 354-356, we wrote:

Higher accuracy of the MWRI-ASI SIC during winter compared to during summer is due to the high sensitivity of PM signal to atmospheric and ice surface melting conditions.

*379/Caption Tab.5: Add "The statistically significant at 95% and 99% confidence levels are marked by \* and \*\*, respectively."*

**Reply:**

We have added it. Due to the comment "213/Caption Fig.1: Correct 'statistical significant' to 'statistical significance'.", we further corrected this sentence.

Page 18, Line 375, we wrote:

The statistical significance at 95% and 99% confidence levels are marked by * and **, respectively.

*390: Replace "more intense." with "stronger".*

**Reply:**

We have replaced it.

Page 19, Line 385-388, we wrote:

In the PIZ, the MADs of polarization difference at 89 GHz between the MWRI TB and SSMI TB are smaller with values of 1.5 and 1.7 K in the Arctic and Antarctic, respectively, compared to those in the MIZ (4.6 and 4.4 K) and over open water (7.6 and 7.6 K), where the atmospheric influence is stronger.

**Reply:**

We have corrected it.

Page 21, Line 423-425 we wrote:

Compared to the SSMI-ASI and AMSR-ASI, the MWRI-ASI reveals more ice along the coasts and around the islands, such as around 72° N from 138° to 144° E in the Arctic and around 78° S from 160° W to 170° E in the Antarctic (Fig. 9).

**Reply:**

We have corrected it.

Page 21, Line 452-454, we wrote:

In the Antarctic, our MWRI-ASI SIE is about two grid cells (25 km) farther south than the Sea Ice Index SIE, and the absolute deviation increases in winter because the SIE is larger in winter than in summer.

*474: Correct "temporary data gaps" to "temporal data gaps".*

**Reply:**

We have corrected it.

Page 23, Line 471-472, we wrote:

This dataset is available from 12 November 2010 to 31 December 2019 with temporal data gaps of 23 days in the Arctic and 82 days in the Antarctic.

*477/478: Change "'NoData' is the missing data." to "'NoData' for missing data."*

**Reply:**

We have changed it.

Page 23, Line 475-476, we wrote:

The values '0-100' are the percentage of SIC, flag of '-1' is the land and of '-2' is the Pole Hole, and 'NoData' for missing data.

**Reply:**

We have replaced it.

Page 23, Line 483-485, we wrote:

To test the skill of the MWRI-ASI as an independent PM SIC dataset, the MWRI-ASI SIC is compared to the existing ASI SIC products of SSMI-ASI and AMSR-ASI, and the MWRI-ASI SIE is compared to the existing SIE products of SSMI-BST, SSMI-NT, OSI-SAF, and Sea Ice Index.

*500: Change "Following this study, we will attempt" to "Future improvements are aimed".*

**Reply:**

We have changed it.

Page 24, Line 498-499, we wrote:

Future improvements are aimed to identify and remove the spurious sea ice caused by land spillover and weather effects more accurately by using satellite-based observations with higher resolutions to further improve the MWRI-ASI SIC.

*522: Suggest to acknowledge the input of the reviewers, pls.*

**Reply:**

We have added it.

Page 26, Line 521, we wrote:

In addition, we are grateful to the reviewers and editors for their improvements to this paper.

*References:*

*Add new refence:"esastar, https://esastar-publication-ext.sso.esa.int/ESATenderActions/details/55123, accessed 2023-0610." (Info accessed -- not to be included in your manuscript: "The Copernicus Imaging Microwave Radiometer (CIMR) mission is an Expansion mission of the European Commission to enlarge the Copernicus Space Component. The aim of CIMR is to provide high-spatial resolution microwave imaging radiometry measurements and derived products with global coverage and sub-daily revisit in the polar regions and adjacent seas to address Copernicus user needs. The aim of this activity is to prepare an airborne capability that is required for CIMR calibration and validation activities both prior and post CIMR-A launch expected in 2028/29.")*

**Reply:**

We have added it according to the requirement of references.

Page 27, Line 560, we wrote:

Esastar: https://esastar-publication-ext.sso.esa.int/ESATenderActions/details/55123, last access: 10 June 2023.